


# Now You See It Now You Don't: A Case Study of Ephemeral Snowpacks in the Great Basin U.S.A.

Rose Petersky[1] and Adrian Harpold[1]

[1]Hydrology, University of Nevada Reno 1664 N Virginia St Reno, NV 89557

*Correspondence to:* Adrian Harpold (aharpold@cabnr.unr.edu)

## 1 Abstract

Ephemeral snowpacks, or those that routinely experience accumulation and ablation at the same time and persist for <60 days, are challenging to observe and model. Using 328 site years from the Great Basin, we show that ephemeral snowmelt delivers water earlier than seasonal snowmelt. For example, we found that day of peak soil moisture preceded day of last snowmelt in

the Great Basin by 79 days for shallow soil moisture in ephemeral snowmelt compared to 5 days for seasonal snowmelt. To understand Great Basin snow distribution, we used moderate resolution imaging spectroradiometer (MODIS) and Snow Data Assimilation System (SNODAS) data from water years 2005-2014 to map snow extent. During this time period snowpack was highly variable. The maximum seasonal snow cover was 64 % in 2010 and the minimum was 24 % in 2014. We found that elevation had a strong control on snow ephemerality, and nearly all snowpacks over 2500 m were seasonal. Snowpacks

were more likely to be ephemeral on south facing slopes than north facing slopes at elevations above 2500 m. Additionally, we used SNODAS-derived estimates of solid and liquid precipitation, melt, sublimation, and blowing snow sublimation to define snow ephemerality mechanisms. In warm years, the Great Basin shifts to ephemerally dominant as the rain-snow transition increases in elevation. Given that snow ephemerality is expected to increase as a consequence of climate change, we put forward several challenges and recommendations to bolster physics based modeling of ephemeral snow such as better metrics

for snow ephemerality and more ground-based observations.

## 2 Introduction

Seasonal snowmelt supplies water to one-sixth of the world's population, which supports one-fourth of the global economy (Barnett et al., 2005; Sturm et al., 2017). Seasonal snowpack provides predictable melt timing and volumes in the spring, which influences streamflow timing, surface water and groundwater availability (Berghuijs et al., 2014; Jasechko et al., 2014;

Stewart et al., 2005). Reliable spring snowmelt also provides a strong control on vegetation phenology and productivity in many ecosystems (Parida and Buermann, 2014; Trujillo et al., 2012). Despite the importance of seasonal snow to water supplies, much of the world's snow is ephemeral, which means it melts and sublimates throughout the snow cover season instead of having one consistent period of snowmelt. Even small shifts from seasonal to ephemeral snowpack due to regional warming could disrupt





snowmelt timing in ways that could alter summer productivity, soil temperature, and soil moisture regimes (Hamlet et al., 2005; Harpold and Molotch, 2015; Jefferson, 2011; Parida and Buermann, 2014; Regonda et al., 2005; Stielstra et al., 2015; Trujillo et al., 2012). A shift from seasonal to ephemeral snowpacks will also have negative implications for the winter tourism, water management, hydropower, and forest management sectors in particular (Schmucki et al., 2014; Sturm et al., 2017). Despite

the hydrological and ecological importance of ephemeral snow, we lack widely accepted methodologies to classify, map, and model snow ephemerality.

One widely accepted snowpack classification system in snow hydrology by Sturm et al. (1995) divides snowpack into six categories: Tundra, Taiga, Alpine, Maritime, Ephemeral, and Prairie. In that system, ephemeral snowpacks are defined as all snowpacks that persist for less than 60 consecutive days, are less than 50 cm depth, and have less than three different

snow layers (Sturm et al., 1995). The Sturm et al. (1995) classification system is also incorporated into physical snowpack models, such as SnowModel (Liston and Elder, 2006), to separate seasonal and ephemeral snowpacks. Models often separate the calculation of seasonal and ephemeral snowpack energetics because ephemeral snowpacks are much more sensitive to basal melt from ground heat flux. Additionally, cold content varies more in shallow ephemeral snowpacks. Although not much is known about their hydrological impacts, ephemeral snowpacks modify the intensity and duration of precipitation inputs by

storing and releasing water in a less predictable way than seasonal snow. While it is arbitrary, using the 60-day threshold allows for comparable estimates of the extent of ephemeral snow and resulting implications of increased snow ephemerality.

Ground-based and remote sensing observations have their own strengths and weaknesses for observing ephemeral snowpacks. Most ground-based snow measurement stations (e.g. the National Resource Conservation Snow Telemetry, NRCS SNOTEL) in the Great Basin–and the rest of the Western United States–are built to observe seasonal snow (Fig. 1). This is because

sites are typically placed in topographically sheltered forest gaps that retain snow longer than nearby terrain. This improves the skill of streamflow forecasting, the primary goal of the SNOTEL network, but means that most SNOTEL sites only have ephemeral snow cover in exceptionally dry or warm years (Serreze et al., 1999). The scarcity of ground-based ephemeral snow data has changed slightly in recent years with additional measurements at NRCS SCAN (Fig. 1) and within research watersheds (Anderton et al., 2002; Jost et al., 2007). However, the lack of field observations from ephemeral snowpacks has limited

previous investigations (e.g. Sturm et al. 2010).

Spectral remote sensing collects observations over all cloud-free areas, including both seasonal and ephemeral snow zones, but has its own sets of advantages and challenges. There are multiple methods to define the start and end of the observed snow covered period. Often, it is defined as the date of the first and last remotely sensed observations of snow cover (e.g. Choi et al. 2010; Kimball et al. 2004; Nitta et al. 2014). Because this approach does not account for intermittent snow free

periods, it tends to overestimate snow duration and miss important ephemeral dynamics Thompson and Lees (2014). Snow persistence thresholds can be used to define snow ephemerality, but no standard persistence threshold exists (e.g. Gao et al. 2011; Karlsen et al. 2007). Given the intermittent nature of ephemeral snow, observations must be daily or finer to capture its dynamics (Wang et al., 2014). Consequently, products like Landsat that have a 16-day overpass do poorly at estimating snow seasonality compared to products like the moderate-resolution imaging spectroradiometer (MODIS). Moreover, high cloud

cover reduces observation frequency, and limits the ability to observe ephemeral snow events. Like with ground-based snow



research, some remote-sensing based studies exclude ephemeral events altogether (e.g Sugg et al. 2014). The algorithm developed by Thompson and Lees (2014) removed most of the methodological flaws mentioned above by using daily MOD10A1 data and accounting for snow absences in the middle of the snow season, but their study was challenging to verify and applied only in a small area of Australia. Given the current lack of ground-based observations (Fig. 1), remote sensing is one path forward for observing ephemeral snow.

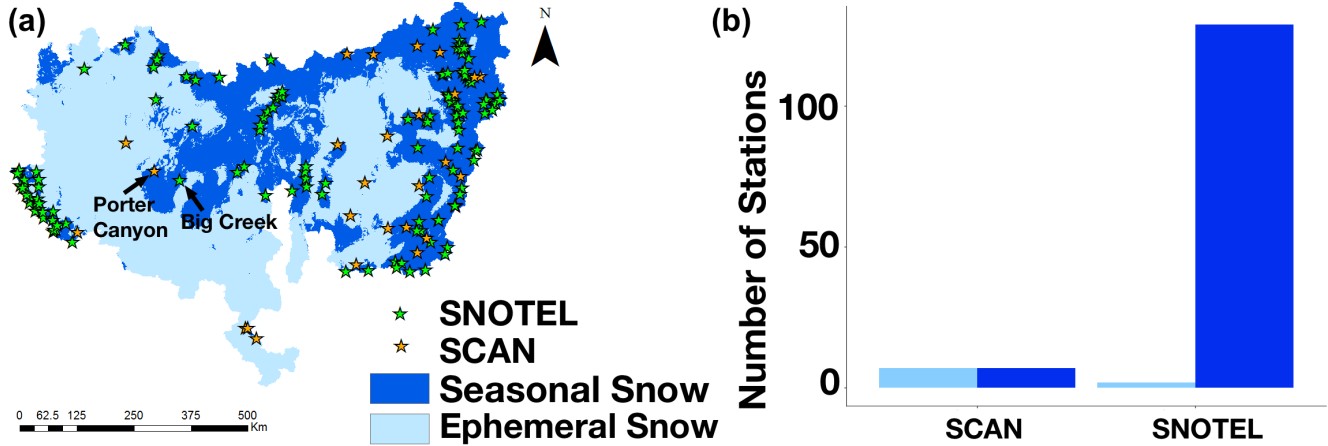

**Figure 1.** (a) Locations of and (b) Number of Snow Telemetery (SNOTEL) and Soil Climate Analysis Network (SCAN) stations in the Great Basin, USA that are located in ephemeral and seasonal snow as defined by <60 or >=60 days of maximum consecutive snow duration respectively. Snow duration data collected using the Snow Data Assimilation System model.

Modeling ephemeral snowpacks is challenging and has not received the same attention as modeling more persistent, seasonal snowpacks. Most physics based models (e.g. Liston and Elder 2006), are optimized for seasonal snow, and produce less accurate results over ephemeral snow (Kelleners et al., 2010; Kormos et al., 2014). As stated previously however, there is a lack of field observations to interrogate and verify these models against Sturm et al. (1995); Toure et al. (2016).

There are a variety of underlying processes that cause ephemeral snowpack and challenge snow models. Based on previous classification systems, we define three mechanisms causing ephemeral snowpacks: 1) Rainfall limiting the accumulation of snowpack, 2) Snowpack ablation from melt or sublimation, and 3) Wind scour removing snowpacks. All of these mechanisms have a variety of underlying atmospheric and snowpack processes that challenge prediction with snow models. At rain-snow transition elevations, even small temperature variations and other atmospheric variables can alter the mixture of rainfall and snowfall (Henderson and Leathers, 2010; Jefferson, 2011; Klos et al., 2014; Regonda et al., 2005). Complete snow water equivalent (SWE) removal from melt or sublimation is also another common cause of snow ephemerality (Clow, 2010; Leathers et al., 2004; Mote et al., 2005; Sospedra-Alfonso and Merryfield, 2017). Typically, physics based models overestimate modeled SWE in ephemeral snowpack, due to neglect or underestimation of ground heat flux and the challenges of tracking cold content in shallow snowpacks (Cline, 1997; Hawkins and Ellis, 2007; Kelleners et al., 2010; Kormos et al., 2014; LaMontagne, 2009; Şensoy et al., 2006). Models parameterize energy fluxes differently, which can lead to differences in model estimates of





sublimation and melt (Essery et al., 2009; Sospedra-Alfonso et al., 2016; Schmucki et al., 2014). Removal of snowpack from wind scour is a very important factor on snow accumulation in alpine regions, but is often neglected in models altogether (e.g. Mernild et al. 2017; Pomeroy 1991; Winstral et al. 2013). Widespread evidence exists that wind redistribution of snow can cause ephemeral snowpacks that are consistent from year to year because of topography and dominant wind directions (Hood

et al., 1999). The three mechanisms causing ephemeral snow (i.e. rain-snow transition, ablation by sublimation and melt, and wind scour) have fundamentally different underlying causes, with different and poorly quantified sensitivities to climate and land cover variability.

The goal of this paper is to use the Great Basin as a case study to estimate the distribution, hydrological consequences, and mechanisms of ephemeral snowpacks using both ground-based and remote sensing observations. We adapt a classification

from Sturm et al. (1995) to map snow across the Great Basin, compare remotely sensed and modeled estimates of ephemeral snow, and develop our own metrics to further classify snow seasonality. The Great Basin is ideal for this investigation because it spans dramatic gradients of elevation and hydroclimatology. This prototypical area depends disproportionately on mountain snowpack for water supplies, contains few ground-based observations, and there is relatively little winter cloud cover to limit spectral remote sensing techniques. Three research questions guide our analyses of ephemeral snowpacks in the Great Basin: 1)

What are the implications for soil moisture from seasonal to ephemeral snow melt? 2) How does topography affect snow seasonality? and 3) What mechanisms cause ephemeral snowpacks and how does that vary with climate? We find that ephemeral snowmelt leads to fundamentally different water availability than seasonal snow that results when melt and rain-snow transition shift lower in elevation during warm winters.

## 3 Study Area

The Great Basin is the closed basin between the Wasatch and southern mountain ranges in Utah and the eastern slope of the Sierra Nevada mountain range in California. The region is known for having "internal drainage," which means that none of the waterways travel to the ocean (Svejcar, 2015). The climate is semi-arid and the ecosystem is shrub-dominated (Svejcar, 2015; West, 1983; Wigand et al., 1995). We defined the Great Basin region based on the Hydrologic Unit Code (HUC) Region 16 adapted from Seaber et al. (1987) by the United States Geological Survey (USGS) (Fig. 2). Precipitation in the Great Basin

varies widely between <10 cm in many of the lower elevations to >100 cm on many of the high elevation mountains (Fig. 3). Overall, the Great Basin has a mean winter precipitation of 12 cm and a mean winter temperature of 0.4 degrees C (Fig. 3) (Abatzoglou, 2012).

## 4 Methods

In order to compare the effect of snow ephemerality on soil moisture patterns, we first investigated snow and soil moisture

response for SNOTEL and SCAN stations within the Great Basin. To evaluate how soil moisture varies based on snowpack parameters during a drought year (water year 2015) and a non-drought year (water year 2016), we chose two SNOTEL stations:



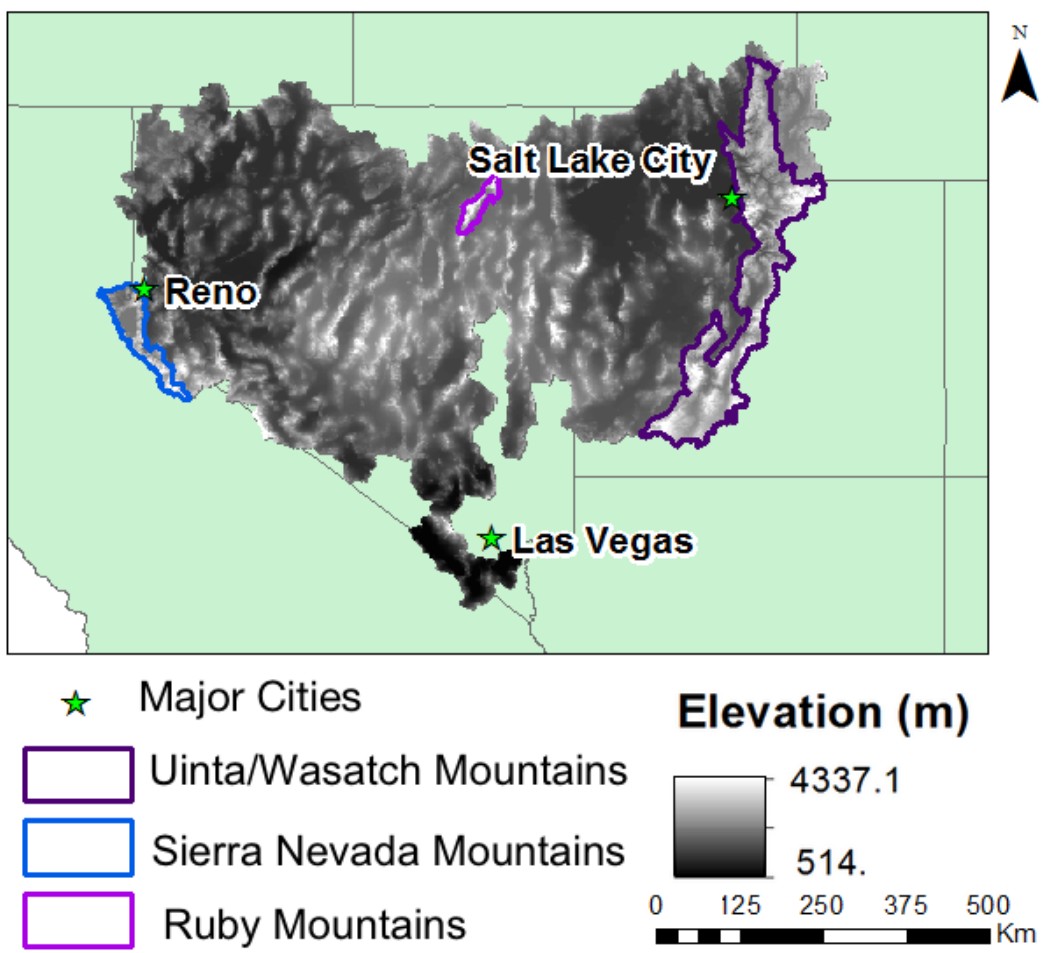

**Figure 2.** Map of the Great Basin region, USA as defined by the United States Geological Survey (USGS) Hydrologic Unit Code (HUC) Region 16 along with major cities and mountain ranges. The Sierra Nevada and Wasatch/Uinta mountain ranges defined using the US EPA L4 ecoregion classifications of "Sierra Nevada" and "Wasatch Uinta" respectively. Ruby Mountains were defined using a combination of "Mid-Elevation Ruby Mountains" and "High Elevation Ruby Mountains" in the US EPA L3 classification Omernik (1987). Elevation contours at 1000 m intervals.

Porter Canyon (ID: 2170, Elevation 2191m) and Big Creek Summit (ID: 337, Elevation 2647m) that differ in elevation but are in close proximity. We then used average snow water equivalent (SWE) data across water years 2005-2014 from the snow data assimilation (SNODAS) model to categorize each SNOTEL and SCAN station as being in ephemeral or seasonal snow if the duration of continuous snow cover was less or greater than 60 days, respectively. For these stations, we compared percent soil moisture, at 5 and 50 cm soil depth along with snow depth, and SWE. We also acquired soil moisture and SWE data at 5 and 50 cm for all the SNOTEL and SCAN stations in the Great Basin in water years 2014-2016. We discarded years and stations



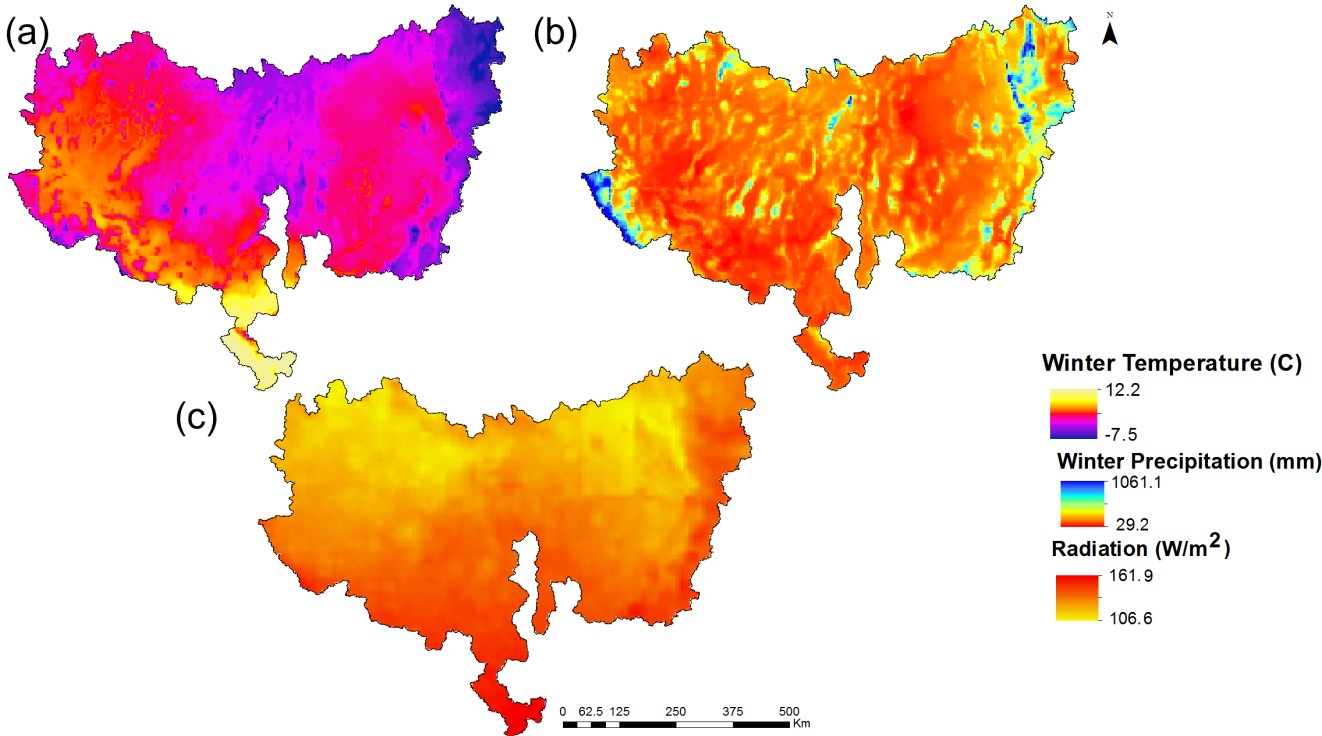

**Figure 3.** (a) Average winter temperature,(b) average winter precipitation, and (c) average winter radiation across water years 2001-2015 along with elevation in the Great Basin.

containing more than seven days of continuous missing data or soil moisture values that were 0 %. To compare the timing of snow and peak soil moisture, we then took the difference between the day of last snow and the day with peak median 10 day soil moisture for each year at each site. We also calculated the coefficient of variation (one standard deviation divided by the mean) of soil moisture for each year at each station. We used the maximum length of continuous SWE that was greater than

5   0.1 cm in to categorize years as containing ephemeral or seasonal snow.

We mapped ephemeral snow across the Great Basin using two methods: spectral remote sensing with MODIS data and SNODAS data. We used Google Earth Engine to analyze the data, which is a cloud-based computing platform optimized for mapping large datasets. The MODIS dataset we used was the 2010 MODIS/Terra Snow Cover Daily L3 Global 500 m Grid (MOD10A) and we used the Normalized Difference Snow Index (NDSI) with parameters outlined in Hall et al. (2006) to find

10   fractional snow covered data. The equation for calculating NDSI in MOD10 is:

$$NDSI = \frac{Band4 - Band6}{Band4 + Band6} \tag{1}$$

A pixel is then mapped as containing fractional snow based on the NDSI value and the percent reflectance value in Band 2. If the reflectance is less than 10 %, the pixel won't be mapped as containing snow regardless of the NDSI value (Hall et al., 2001).





We classified all pixels with a snow fraction of 30-100 as Snow, pixels with snow fractions between 0 and 30 as No Snow, and pixels that had all other designations as Other. We also used an algorithm derived from Sturm et al. (1995) to minimize the impact of cloud cover in our MODIS data. The algorithm 'grows' the boundaries of all areas containing snow and reclassifies pixels that were classified as Other to Snow if the corresponding pixels in the previous image were classified as Snow. It also

reclassifies pixels that were classified as Other to No Snow if the corresponding pixels in the previous image were No Snow.

To determine the number of ephemeral and seasonal snow events, we used a Google Earth Engine function to note the day of the water year when snow appeared (when a pixel went from classified as No Snow in the previous day to classified as Snow in the current day) and when snow disappeared (a pixel went from classified as Snow in the previous day to being classified as No Snow in the current day), and determined the length of snow cover by subtracting the day of snow appearance from the day

of snow disappearance. If the length of snow cover was <60 days, then the snow event was classified as ephemeral. Otherwise, if the length of snow cover was ≥60 days, the snow event was categorized as seasonal. In addition to these metrics, we derived a snow seasonality metric (SSM) to quantify a MODIS pixel's tendency to have ephemeral or seasonal snow, rather than a binary metric like <60 days. The SSM is depicted in Eq. 2 and it works by classifying every day where there was seasonal snow present as 1 and every day where there was ephemeral snow present as -1, and then averaging all -1 and +1 values. This

created a -1 to 1 scale, where -1 signifies that all the snow covered days in a given pixel within one water year were ephemeral and +1 signifies that they were all seasonal.

$$SSM = \frac{Days_{Seasonal} - Days_{Ephemeral}}{Days_{Total}} \qquad (2)$$

Additionally, we discarded all instances where snow was absent for one day only from the overall record of snow disappearance and appearance because we found numerous artifacts from the MOD10A NDSI processing that lead to single day snow

disappearance during long stretches of snow cover. One day snow events were also removed from the SNODAS algorithm to make both algorithms more consistent. For each water year from 2001 to 2015, we recorded the maximum total number of days where snow was present (to be referred to as the maximum snow duration).

To determine the relationship between elevation and snow seasonality, we took the average maximum snow duration across water years 2001-2015 and used elevation, and aspect as measured by a digital elevation model (DEM) obtained from the

Shuttle Topography Mission resampled to the same resolution with bilinear sampling (Farr et al., 2007). To calculate northness, we used the equation:

$$Northness = cos(\frac{aspect * \pi}{180}) \qquad (3)$$

We then categorized each MODIS pixel based on five 500 m elevation bins from a range of 1000 to >3000 m. Then, to remove bias based on the size of each bin, we used random sampling to make each bin contain the same number of points

as the least full bin (13548 points that were >3000 m). Then we combined each resampled bin into one dataset and created heatmaps to compare the elevation vs. the average maximum snow duration. We also use the same method to compare aspect



to average maximum snow duration with aspect using eight 45 degree bins from a range of 0 to 360 degrees. We randomly sampled 195163 points from each bin (the size of the bin from 315 to 360 degrees). After resampling, we combined all the bins together and split them into three elevation categories: Low Elevation (Elevation <1500 m), Medium Elevation (1500 ≥ Elevation <2500), and High Elevation (Elevation ≥ 2500m). Then, we resampled again to 82823 points per bin (the size of the

5   High Elevation bin).

We used SNODAS data to differentiate the mechanisms that cause snow to become ephemeral. The four mechanisms were assigned if the net ablation (or rain) exceeded 50 % of the total winter precipitation (Fig. 4): 1) A mixture of rain and snow limiting snow accumulation (the rain-snow transition), 2) snowpack loss due to sublimation, 3) snowpack loss due to melt, and 4) snowpack loss due to wind scour. We determined the prevailing mechanism in each 1000 m SNODAS pixel in each year.

10   We used Earth Engine to execute the modeled algorithm on each 1000 m SNODAS pixel in the Great Basin. We then chose six years (2009-2014) and created histograms of each mechanism by elevation for each year.

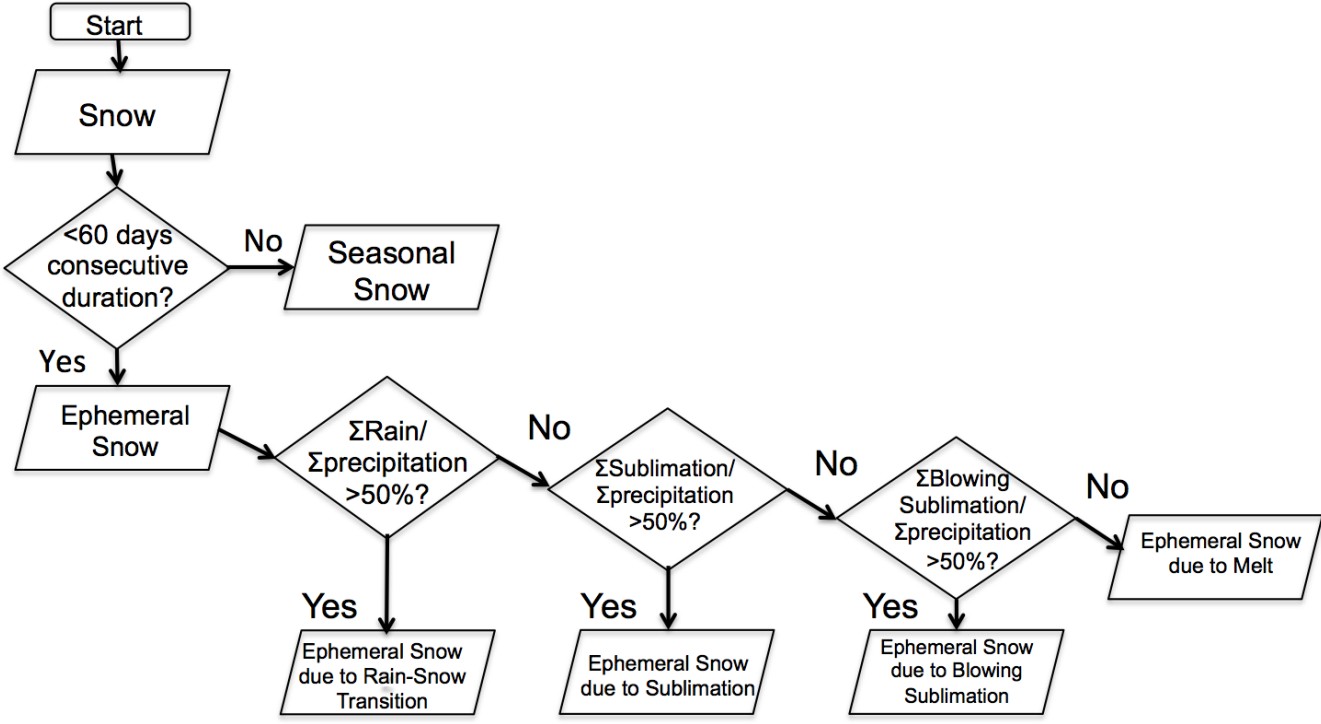

**Figure 4.** Diagram of the process for the ephemeral snow mechanism model. Seasonal snow outputs were rejected, all other outputs were categorized.



## 5   Results and Discussion

### 5.1   Ephemeral Snow and Soil Water Inputs

Snowmelt influences a variety of terrestrial hydrological processes and states, but it has a dominant influence on infiltration and soil moisture dynamics in areas with low summer precipitation (Harpold and Molotch, 2015). Soil moisture is a primary
control on rainfall-runoff response and water availability for vegetation (McNamara et al., 2005; Schwinning and Sala, 2004).

We quantified differing soil moisture responses between seasonal and ephemeral snowpacks that have important ecohydrological implications. McNamara et al. (2005) described five phases of soil moisture evolution in semi-arid watersheds with seasonally dominant snowmelt: (1) a summer dry period, (2) a transitional fall wetting period, (3) a winter wet, low-flux period, (4) a spring wet, high-flux period, and (5) a transitional late-spring drying period. We use the McNamara et al. (2005)
framework for soil moisture response to seasonal snowmelt to illustrate differences with soil moisture response to ephemeral snow melt. First, by qualitatively using two nearby sites with differing snow regimes. Then second, by quantitatively using all of the soil moisture records available in the Great Basin (Fig. 5).

We contrast soil moisture response at two adjacent SNOTEL stations that differ in elevation by >500 m (Fig. 1) to illustrate differences between ephemeral and seasonal snowmelt. Soil moisture at 5 and 50 cm was used to represent shallow and deep
responses during a drought year (water year 2015) and a typical year (water year 2016). Porter Canyon had ephemeral snow (28 days maximum duration) in 2015 and seasonal snow (116 days) in 2016 (Fig. 5a). Big Creek had seasonal snowpack both years, although much shallower snowpack in 2015 (Fig. 5b). When seasonal winter snowpack is present at both sites in 2016, soil moisture follows the phases outlined by McNamara et al. (2005) for a semi-arid, snowmelt driven environment during seasonal snowpack in 2016. Shallow and deep soil moisture was in a low-flux state during December-February (DJF) at Big
Creek in 2016 (Fig. 5f). During March-May (MAM), soil moisture increased substantially and was in a high-flux state. Average shallow soil moisture was similar in the MAM period (24.4 % and 24.8 %, respectively) and DJF period (11.3 % and 19.8 %) between 2015 and 2016, suggesting that snow storage and melt negates differences in early season soil moisture between years with very different winter precipitation. Porter Canyon also showed a similar soil moisture increase in the MAM period after a stable low-flux pattern in the DJF period during water year 2016. Both sites also reach their near maximum annual soil
moisture coincident with snow (Harpold and Molotch, 2015) in 2016, but Porter Canyon has snow disappearance in both years that preceded peak soil moisture by several months. The deeper 50 cm soil moisture had a smaller and shorter peak during 2015 at Porter Canyon as compared to 2016 and Big Creek response.

In addition to comparing soil moisture responses for two sites, we also analyzed 328 site years (50 ephemeral and 278 seasonal site years) from all SNOTEL and SCAN sites in the Great Basin (Fig. 1) over water year 2014, 2015, and 2016 in
order to illustrate the broader patterns of soil moisture between ephemeral and seasonal snow melt. We found that soil moisture following seasonal snow melt peaked on average 5 and 7 days prior to snow disappearance for shallow and deep soil moisture, respectively. This confirms previous findings that seasonal snow melt drives coincident wetting and deeper water percolation (Harpold and Molotch, 2015; McNamara et al., 2005). In contrast, the median soil moisture peaked 79 and 48 days after of snow disappearance from ephemeral snow melt for shallow and deep soil moisture, respectively (Fig. 6a). Deep soil moisture in



## Porter Canyon    Big Creek

**Figure 5.** (a,b) Snow depth, (c,d) Snow Water Equivalent and (e,f) Soil Moisture measured at Porter Canyon and Big Creek Snow Telemetry (SNOTEL) stations for water years 2015-2016, which were a drought year and a typical year respectively.

ephemeral snowmelt had a coefficient of variation (CV) of 0.2 compared to 0.4-0.5 for seasonal snowmelt (Fig. 6). The lower CV for deep ephemeral snowmelt compared to deep seasonal snow melt likely reflects reduced deep percolation becoming available to groundwater and streamflow.

The differences in soil moisture response between seasonal and ephemeral snowpacks across the Great Basin could have
5  important consequences for vegetation phenology and runoff generation. For example, the timing of soil moisture is a strong control on the timing and amount of net ecosystem productivity (Inouye, 2008), with earlier snowmelt causing an earlier



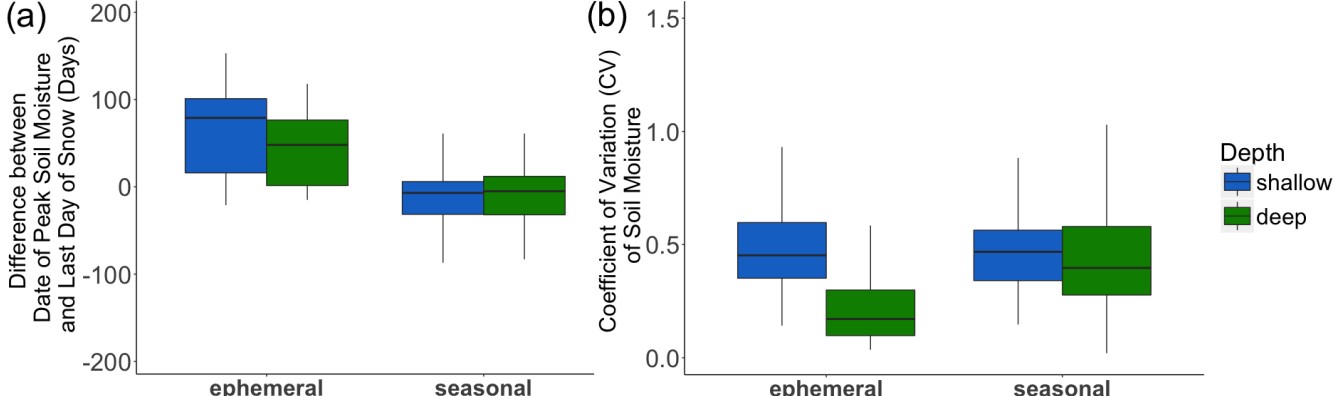

**Figure 6.** (a) The difference between date of peak soil moisture and last day of snow (Days) for shallow (5 cm) and deep (50 cm) soil moisture during water years 2014-2016 in Great Basin SNOTEL stations with ephemeral snow (50 years) and seasonal snow (278 years). (b) The coefficient of variation (CV) for shallow (5 cm) and deep (50 cm) soil moisture during water years 2014-2016 in Great Basin SNOTEL and SCAN stations is also shown.

and longer growing season with reduced carbon uptake (Hu et al., 2010; Winchell et al., 2016). Harpold (2016) also showed that earlier snow disappearance generally led to more days of soil moisture below wilting point at SNOTEL sites across the Western U.S. Our finding that soil moisture peaked earlier in ephemeral snow melt than seasonal snowmelt is thus likely to be correlated with reduced vegetation productivity and increased late season water stress in many areas. In addition to stressing

5   local vegetation, ephemeral snowmelt may reduce groundwater recharge and streamflow. For example, baseflow contributions to streamflow and overall water yield declined when snowmelt rates were smaller (Barnhart et al., 2016; Earman et al., 2006; Trujillo and Molotch, 2014), and overall water yields were lower in basins receiving more rain and less snow (Berghuijs et al., 2014). Changes in percolation patterns also affect the distribution of more shallow rooting plants versus deeper rooting plants that need long duration moisture pulses to grow and reproduce (Schwinning and Sala, 2004). These differences in how

10   ephemeral versus seasonal snowmelt affects soil moisture provide a strong motivation to understand the distribution and causes of ephemeral snowpacks across the Great Basin.



## 5.2 Topographic Controls on Snow Seasonality

In a typical year, much the Great Basin experiences ephemeral snow (Fig. 7) that can only be comprehensively observed with remote sensing platforms because of the lack of standard ground stations (Fig. 1) . Using MODIS imagery, we employ two new metrics to estimate snow ephemerality with daily snow cover products: 1) The maximum consecutive snow duration and

2) The snow seasonality metric or SSM. The SSM describes both the consecutive snow season length and shoulder-season ephemerality. A SSM value <1 means an area experiences at least one ephemeral snow event. The average SSM was -0.4 in the Great Basin (Fig. 7). Maximum consecutive snow duration can be compared to the Sturm et al. (1995) 60-day threshold for ephemeral snow, but it is flexible enough to include a threshold of any day length. The average maximum consecutive snow duration in the Great Basin from MODIS data was 42.1 days (Fig. 7). We found slightly different estimates of the

average maximum consecutive snow duration measured using SNODAS of 62.9 days and the average snow seasonality metric (SSM) was -0.4 (Fig. 7). Although SNODAS ephemerality estimates were very similar to MODIS, SNODAS over estimated snow duration and does not capture the elevation caused patterns (Fig. 7). The results of both metrics and both snow datasets are consistent an area that experiences mostly ephemeral snowpacks but contains areas of persistent seasonal snow at higher elevations (Fig. 7).

We investigate elevation and aspect as proxies for snowpack mass and energy dynamics in order to expand our understanding of snow ephemerality beyond mapping. Elevation is a primary control on near surface air temperature due to the adiabatic lapse rate (Bishop et al., 2011; Greuell and Smeets, 2001; Nolin and Daly, 2006). Prior research has found that there is a strong elevation dependence on snowmelt timing, runoff generation, snow water equivalent (SWE), and snow season length (Hunsaker et al., 2012; Jefferson, 2011; Jost et al., 2007; Molotch and Meromy, 2014). Elevation effects likely a summation of variety

of factors, including temperature controls on the rain-snow transition, longwave radiation in cloudy areas, and sensible heat flux. Aspect is often a secondary control on snow distributions because it influences incoming shortwave radiation (Jost et al., 2007; Pomeroy et al., 2003) and wind patterns (Knowles et al., 2015; Leathers et al., 2004; Winstral et al., 2013). Shortwave radiation is the primary driver of ablation via melt and sublimation (Cline, 1997; Marks and Dozier, 1992).

Splitting the Great Basin into low elevations (<1500 m), mid elevations (1500-2500 m), and high elevations (>2500 m)

illustrated the dominant role that elevation has on snow cover duration (Fig. 8). In our area-normalized sample, 96.2 % of low elevation area and 75.2 % of mid elevation area had a maximum consecutive snow duration of less than 60 days. Only 10.5 % of high elevations had a maximum consecutive snow duration of less than 60 days (Fig. 8). The results suggest that mid and low elevations of the Great Basin are more likely to be ephemerally dominant. The heat maps illustrate that elevation alone is not a strong predictor of maximum consecutive snow cover days (Fig. 8). We use three smaller ecoregions that are focused

on three distinct mountain ranges (see Fig. 2) to illustrate variability in elevation effects (Fig. 9). There were similar average maximum snow duration values in the Ruby Mountains (Fig. 9a), eastern Sierra Nevada (Fig. 9b), and western Wasatch/Uinta ecoregion (Fig. 9c) (107, 100, and 95 days, respectively). However, the Ruby Mountains tended to have longer persisting snow than the Sierra Nevada and Wasatch/Uinta ecoregions. The Sierra Nevada ecoregion had a weaker relationship between snow persistence and elevation above 2500 m, while the Wasatch/Uinta ecoregion had a weaker relationship with elevation below



**Figure 7.** Average maximum consecutive snow duration (maximum snow duration) and snow seasonality metric (SSM) for the Great Basin measured using moderate resolution imaging spectroradiometer (MODIS) and snow data assimilation system (SNODAS) data in the Great Basin, USA. MODIS data is from water years 2001-2015 and SNODAS data is from water years 2005-2014.

2500 m (Fig. 9). These differing relationships between maximum snow duration and elevation point to other factors affecting snow ephemerality.

Aspect is also an important control on snow seasonality in the Great Basin, but its importance is limited to mid and high elevations. We find that there are shorter maximum snow durations in south-facing aspects at elevations >1500 m (Fig. 10). At low elevations, the difference in average maximum snow duration between north and south facing slopes was 0.4 days, while for mid and high elevations, it was 2 and 5 days, respectively (Fig. 10). This is consistent with aspect being a control of solar radiation, which is the main energy input to the snowpack. This suggests that deeper, high elevation snowpacks ablate in response to greater solar radiation and corresponding warmer temperature on south facing hillslope (Hinckley et al., 2014;





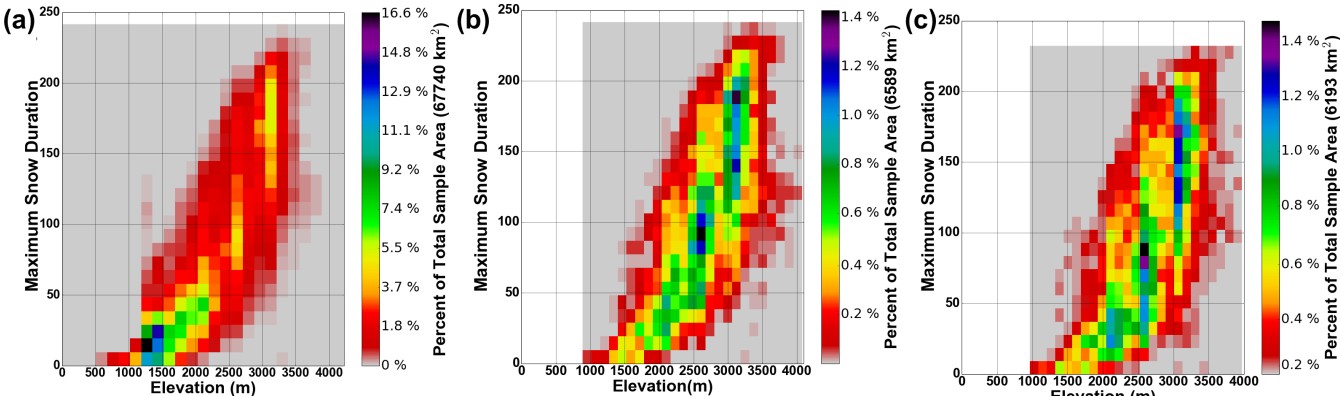

**Figure 8.** Heatmaps of the relationship between elevation and average maximum consecutive snow duration at (a) all slopes,(b) north-facing slopes only, and (c) south facing slopes only in the Great Basin, USA. North facing was defined as Northness >0.25 and south facing was defined as Northness<-0.25. Average maximum snow duration data obtained from moderate-resolution imaging spectroradiometer (MODIS) data, elevation and northness from Shuttle Topography Mission data.

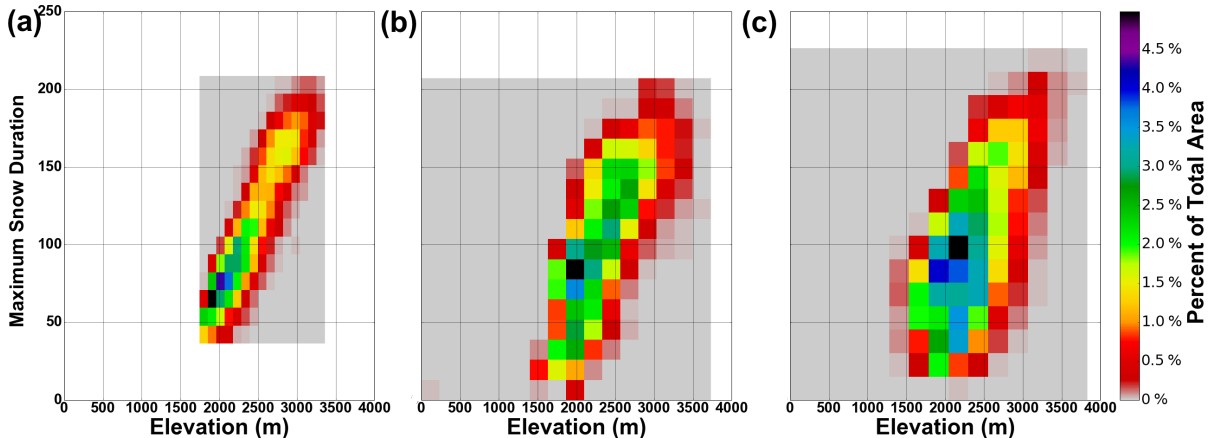

**Figure 9.** Heat maps showing the relationship between elevation and average maximum snow duration for three seasonally-dominant ecoregions in the Great Basin: (a) The Ruby mountains, (b) the Sierra Nevada mountains, and (c) the Wasatch/Uinta mountains.

Kormos et al., 2014). In contrast, lower elevation areas appear to have maximum snow duration caused by factors other than aspect. This is consistent with the outsized importance of other energy fluxes and factors, like ground heat flux and rain-snow transition elevation, that are not captured with simple topographic relationships (Fig. 8, 9 and 10).





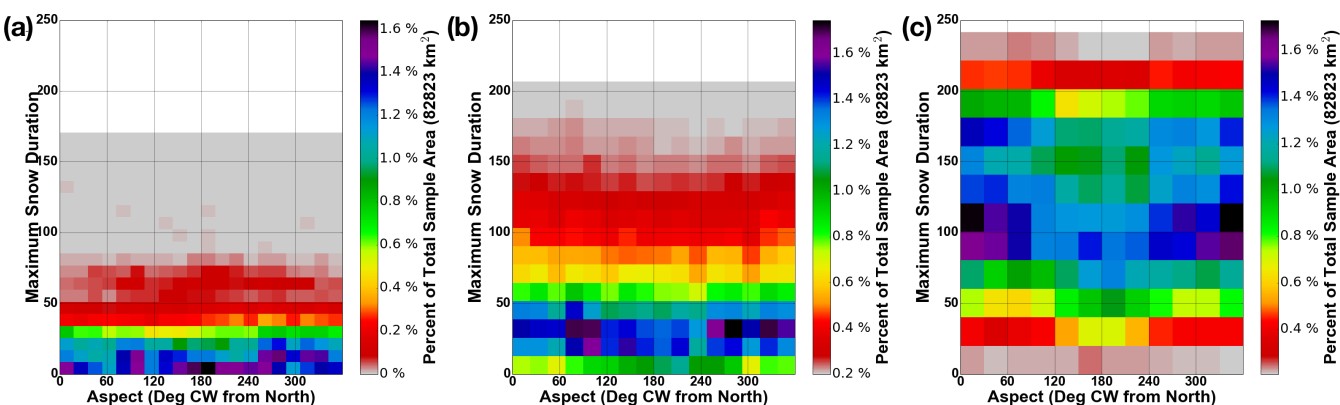

**Figure 10.** Heatmaps of the relationship between aspect and average maximum consecutive snow duration at (a) low elevations (0-1500 m), (b) medium elevations (1500-2500 m) and (c) high elevations (2500 m+).Average maximum snow duration data obtained from moderate-resolution imaging spectroradiometer (MODIS) data, aspect from Shuttle Topography Mission data.



## 5.3 Proximate Mechanisms Controlling Snow Ephemerality

Deciphering the mechanisms controlling ephemeral snowpacks and their sensitivity to climate is challenged by a lack of models and observations. However, we propose a three-mechanism classification scheme to help frame our understanding of snow ephemerality: 1) rain-snow transitions limit snow accumulation, 2) snowpack ablation from melt and sublimation, and 3) wind scour or redistribution. Probably the most explored and observed mechanism is the potential for rising rain-snow transition elevations to limit snow accumulation and duration (Bales et al., 2006; Klos et al., 2014; Knowles and Cayan, 2004; Mote, 2006). Reduction in snow duration can also be caused by the melt of snowpack (Mote, 2006) and losses from sublimation (Harpold et al., 2012; Hood et al., 1999); however, much less is known about the role and distribution of these processes outside of the seasonal snowpack zone. Finally, wind scour can reduce snowpacks by redistributing it to other areas or by increasing blowing wind sublimation (Knowles et al., 2015; Leathers et al., 2004).

We chose six years to evaluate the dominant mechanisms causing snowpack ephemerality using a new classification system (Fig. 3) based on SNODAS data that compared favorably to estimates from MODIS (Fig. 7). In that six year period, the year with the lowest average winter temperature using GRIDMET estimates was 2013 at -0.9°C while the year with the highest average winter temperature was 2014 at 1.0°C (Abatzoglou 2012; Table 1). In water year 2013 and water year 2010, the two coldest years, seasonal snowpacks were dominant in most of the Great Basin and Western United States (Fig. 11-12). In the coldest years of 2010 and 2013, the rain-snow transition and melt caused ephemerality to shift lower in elevation (Fig. 12). In the warmest year (2014), seasonal snowpack was lowest at lower elevations in allthroughout the Western US mountain ranges (Fig. 11)., including the Great Basin where ephemeral snowpacks increased in middle and higher elevations due to the rain-snow mechanism (Fig. 11 and 12). Melt caused snow ephemerality also increased in the warm 2014, but ephemerality ephemeral snow remained low sparse above 2500 m in all years. Sublimation was only present as a limiting mechanism in 2010 and only for a small area (Fig. 11). Blowing snow sublimation was not the dominant cause of snow ephemerality in the Great Basin for any year, but its known that SNODAS struggles to represent wind redistribution of snow Clow et al. (2012); Hedrick et al. (2015).

**Table 1.** Average winter temperature (° C) and average elevation (m) for both dominant mechanisms of snow ephemerality and seasonal snow from 2009-2014 in the Great Basin.

| Water Year | Average Winter Temp (deg C) | Mean Elev for Rain Snow Transition (m) | Mean Elev for Melt (m) | Mean Elev for Seasonal Snow (m) |
|---|---|---|---|---|
| 2009 | 0.1 | 1806.3 | 1750.8 | 1728.4 |
| 2010 | -0.6 | 1811.3 | 1747.1 | 1761.3 |
| 2011 | -0.2 | 1803.7 | 1765.6 | 1699.6 |
| 2012 | 0.4 | 1803.7 | 1745.2 | 1709.8 |
| 2013 | -0.9 | 1815.6 | 1709.8 | 1754.1 |
| 2014 | 1.0 | 1789.9 | 1748.9 | 1731.5 |



The mechanisms causing snow ephemerality that can be inferred from the SNODAS model have important implications for water availability in the Great Basin, but we lack confidence in the model fidelity in these shallow snowpacks. These limitations are present in all snowpack energy models because the models were developed for deeper snowpacks where terms like ground heat flux and albedo-depth relationships can be ignored (Cline, 1997; Harstveit, 1984; LaMontagne, 2009; Liang et al., 1994). In shallow snowpacks, these terms are more critical (Hawkins and Ellis, 2007; LaMontagne, 2009; Şensoy et al., 2006), and the lack of SWE means the internal energy state of the snowpack (i.e. cold content) is more easily varied by short term climate forcing (e.g. warm, sunny days) (Liston, 1995) and thus, more critical to accurately track. Ephemeral snowpacks also exist at lower elevations with warmer soils and increased ground heat flux (LaMontagne, 2009). Uncertainty in the rain-snow transition principally arises from predicting climate forcing and in particular temperature. However, the underlying phase prediction method and related model decisions and climate forcing data can also be important for the quality of precipitation phase prediction (Harpold et al., 2017). Further complicating rain-snow transition mechanisms is storage or drainage of liquid water in the snowpacks (Lundquist et al., 2008; Marks et al., 2001). Although SNODAS assimilates MODIS imagery into the model, it does not appear to capture the finer elevation patterns we found using the MOD10A product (Fig. 7), which is consistent with challenges reported by other SNODAS verification efforts in complex terrain (Clow et al., 2012; Hedrick et al., 2015). The Great Basin shows tremendous sensitivity to snow ephemerality from topography and elevation (Fig. 11-12) and thus, represents an area where improvements in the physically-based modeling of shallow snow and rain-snow transition elevations will be critical to predicting snow water resources under a variable and changing climate.





**Figure 11.** Dominant mechanisms for snow ephemerality from water years 2009-2014 in the Western United States. Data obtained from the snow data assimilation system (SNODAS) model. Areas with seasonal snow, no snow, and water bodies are also depicted. The Great Basin region is outlined in yellow.



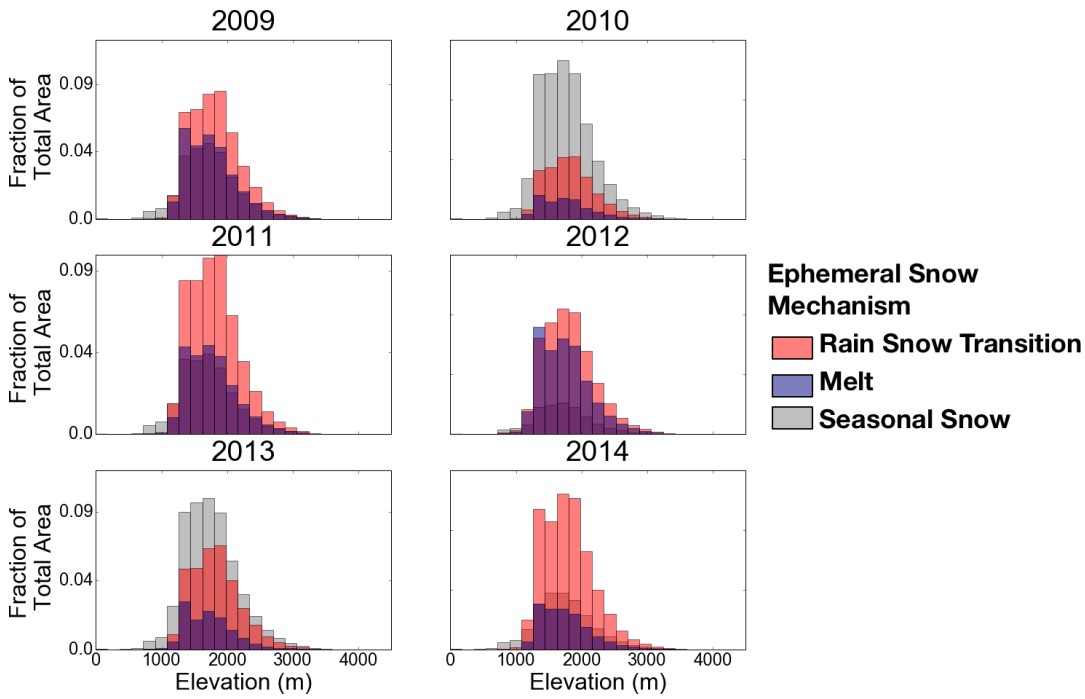

**Figure 12.** Histograms of the relationship between elevation and the dominant mechanisms for snow ephemerality in the Great Basin from water years 2009-2014. Snow data obtained from the snow data assimilation system (SNODAS) model.



## 6 Conclusions

Mapping, measuring, and modeling ephemeral snow is challenging with current techniques, but will be vital for understanding how water resources and vegetation will respond to future climate. Ephemeral snowpacks do not have distinct accumulation and ablation periods, which means the timing of soil moisture input varies and is more challenging to predict. Consequently, as

snowpacks shift from seasonal to ephemeral, there are potential ecohydrological consequences such as changes to vegetation response, vegetation distribution, lateral flow, and solute transport. Our work shows that while topography and climate variability have strong controls on the distribution of ephemeral snowpacks (Fig. 8 and 11), those factors will not be sufficient for predicting snow ephemerality. Instead, we will need physics-based models capable of capturing the three broad mechanisms identified by this study: 1) rain-snow transitions limit snow accumulation, 2) snowpack ablation from melt and sublimation,

and 3) wind scour and redistribution. These classifications could help better identify local and regional sensitivity to increased snow ephemerality (Fig. 11 and 12). This work has also highlighted major weaknesses in the observational infrastructure, data analysis, and modeling techniques needed to support the growing importance of ephemeral snowpacks. In light of these diverse needs, we conclude with a short summary of recommendations meant to guide future directions into this important research topic:

• Better and standard snow ephemerality metrics: Our research suggests there is a snow duration threshold where snowpack and soil moisture patterns begin to resemble seasonal instead of ephemeral snowmelt, and perhaps a second threshold when they begin to resemble rain. Yet evidence that this threshold is the 60 days used in the Sturm et al. (1995) paper is lacking. Instead of using this arbitrary threshold, we recommend that future research use the snow properties and soil moisture response of ephemeral snowpacks combined with a sensitivity analysis to create a snow duration threshold capable of differentiating

seasonal snow melt caused soil moisture (e.g. McNamara et al. 2005) from ephemeral effects and rain.

        • More snow and soil moisture observations in ephemeral areas: In the Great Basin, only two snow telemetry (SNOTEL) stations and 26 soil climate analysis network (SCAN) stations observe ephemeral snowpacks (Fig. 1). The lack of observations makes it more difficult to leverage the clear differences in SWE, snow depth, and shallow soil moisture between ephemeral and seasonal snow. To help develop better criteria for categorizing snowpack as ephemeral, we need more snow and soil moisture

observations in ephemeral areas. We can then also use these observations to verify results derived from remote sensing and physically-based models.

        • Improved remote sensing algorithms: There is currently no consistent standard for defining the length of snow covered periods. It is still common for papers to define the length of a snow covered period by the first and last days of snow cover. This standard does not account for ephemeral events between those days. Additionally, there is no consistent algorithm for

accounting for cloud cover. More widespread use of the object-oriented techniques used in this study is needed to evaluate its efficacy and accuracy across multiple regions.

        • Improved spatial resolution and fidelity of snow and climate data: The MOD10A data product has a spatial resolution of 500 m. The coarse resolution made it difficult to verify our ephemeral snow results with SNOTEL observations that use  3 m wide snow pillows. Topographic complexity leads to variations in climate on much finer resolutions than the 4000 m gridded





meteorology data used for this analysis. Gridded snow and climate data should have a spatial resolution more consistent with the variability in snowpacks on the order of 10-100 meters.

• Improved physics-based modeling: Identifying weaknesses in physically-based models was not the objective of this study; however, it is clear this is a need for better prediction of snow ephemerality. Improving model parameterization of ground heat

5  flux and ensuring the temporal model resolution is sufficient to capture rapid changes in cold content are two ways to improve these models.

## Appendix A: Supplemental Information

**Contents of this file:** Figures A1-A2

**Introduction:**

10  The following figures provide additional information about the ephemeral snow algorithm. Figure A1 shows how the measured number of ephemeral and seasonal snow events at SNOTEL sites corresponded to the number derived from the ephemeral snow algorithm. Figure A2 shows how the 30 % snow fraction was chosen using a sensitivity analysis.





**Figure A1.** Root Mean Square Errors between the number of observed ephemeral and seasonal snow events at Snow Telemetry (SNOTEL) stations and the number of ephemeral and seasonal snow events derived from the algorithm in Google Earth Engine in each 500m Moderate-resolution imaging spectroradiometer (MODIS) pixel corresponding to that station. Measured SWE (Snow Water Equivalent) of 0.3in. or greater was used to determine snow presence for SNOTEL sites.





# Ephemeral Events

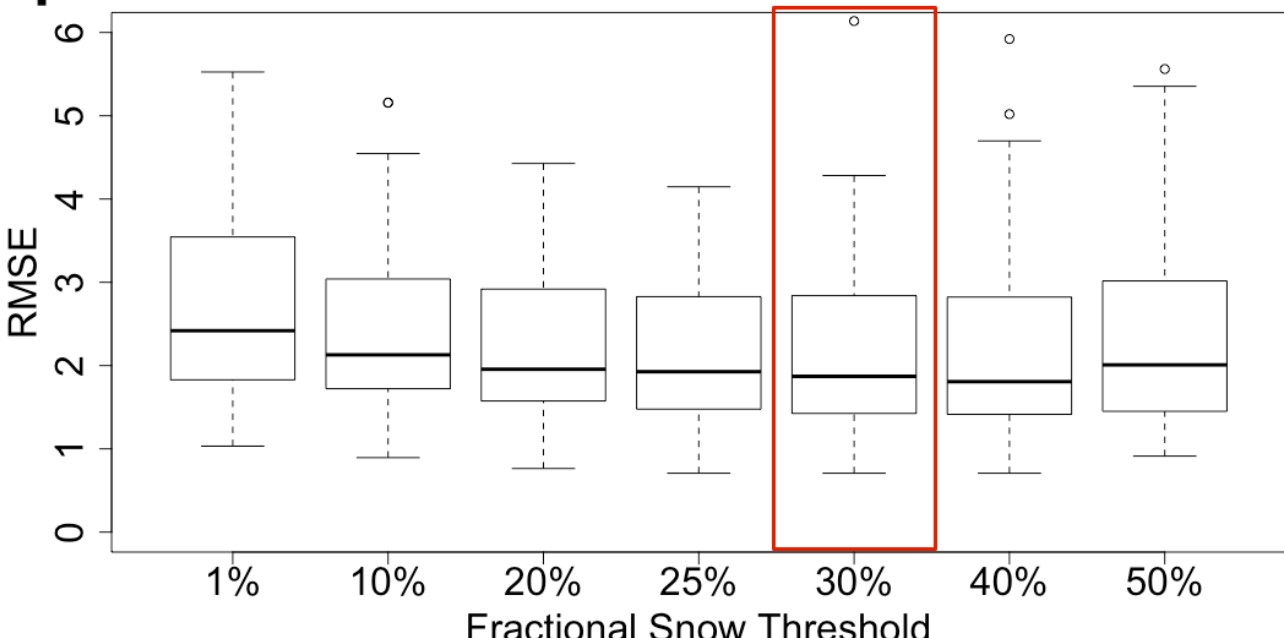

# Seasonal Events

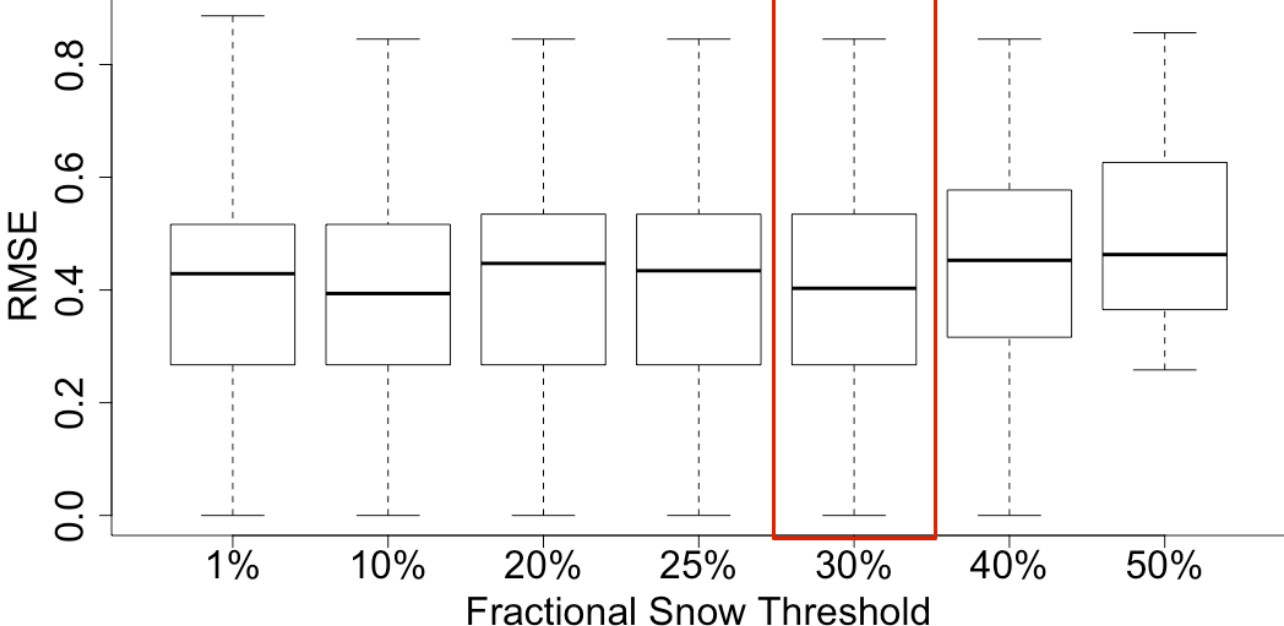

**Figure A2.** Boxplots depicting the Root Mean Square Errors between the number of observed ephemeral and seasonal snow events at Snow Telemetry (SNOTEL) stations and the number of ephemeral and seasonal snow events derived from the algorithm in Google Earth Engine in each 500m Moderate-resolution imaging spectroradiometer (MODIS) pixel corresponding to that station at snow fractions of 1-50 %. 30 % (highlighted in red) was the chosen snow fraction.



*Competing interests.* The authors declare that they have no conflict of interest.

*Acknowledgements.* The NASA Space Grant Consortium and USDA NIFA NEV05293 for providing funding. Patrick Longley helped in creating the snow ephemerality metric. Charles Morton of the Desert Research Institute and the members of the Google Groups Earth Engine Forum helped with Google Earth Engine. We also thank Dr. Scott Tyler for his support.



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
