# Peer review of "Now You See It Now You Don't: A Case Study of Ephemeral Snowpacks and Soil Moisture Response in the Great Basin U.S.A."

_Hydrology and Earth System Sciences, 2017_

## Referee Comment (RC1) · Anonymous Referee #1 · 14 Mar 2018

General comments:

This study focuses on mapping ephemeral snow over the Great Basin in the US and diagnosing the mechanisms for ephemeral snow behavior, to better understand the impacts of ephemeral snow on soil moisture. To do so, the authors use station based SNOTEL and SCAN observations of snow and soil moisture, remotely sensed snow-cover from MODIS, and modeled snow data from the SNODAS product. A snow sea-sonality metric is developed using MODIS snowcover imagery and SNODAS data to map ephemeral and seasonal snowcover. Then a decision tree is developed using SNODAS modeled data to diagnose the mechanism of ephemeral snow. Finally, the

landscape characteristics of estimated snow duration are explored. The study's results are that topography and climate are strong controls on the distribution of ephemeral snowpack. This paper extends previous work on ephemeral snow by attempting to classify the processes driving snow loss (no snowfall, melt, sublimation/blowing snow).

This paper addresses a previous unaddressed question of identifying and mapping the dominate process causing ephemeral snow cover. Further, connecting snowmelt from ephemeral snow to soil moisture and this link's sensitivity to inter-annual variability (or climate change), is an interesting scientific issue fully within the scope of HESS. I believe this paper will provide a valuable contribution after some of the issues below are addressed.

Specific comments:

- Title: The title should be more descriptive of what the paper is about: diagnosing ephemeral snow mechanisms and impacts on soil moisture.

- The abstract provides some descriptive comments on the seasonality of the observed and modeled results, but only hints at "recommendations to bolster physics based modeling". These recommendations (and results supporting them) should be clearly articulated in the abstract.

- Lack of discussion of how uncertainty in the SNODAS model affect the results of this study, namely the classification of ephemeral snow. Over high-elevation terrain where we could expect blowing snow redistribution and sublimation losses to be greatest, SNODAS at 1km by 1km, likely does not capture these processes well. This may be supported by Figure 7, showing SNODAS diverging from MODIS at highest elevations (This is an interesting finding that could be discussed more as well).

- Because ephemeral snow occurs during short events, the driver of snow loss for a given 1km SNODAS cell could be variable with time. How does your ephemeral snow mechanism modeled results change if you look at smaller time slices than a year?

[Figure]

- Snowpillows modify the ground heat flux to snow and the calculated snow presence/absence. Please address how this observational uncertainty impacts your results.

- Using the peak of (I assume hourly?) soil moisture data for your calculations for Figure 6, may bias this metric toward high intensity rainfall events (i.e. Feb 2015 in Figure 5e), that may be slightly higher than later snowmelt driven soil moisture increases. Try using a longer averaging time or at least address the sensitivity of your results to this metric choice.

- Making your final mapped snow regions publicly available will greatly improve the usefulness of this study.

Technical corrections:

- Page 2, 5 – Missing citation of Kormos et al., 2014

- Page 2, 14 – Inputs "to soil"

- Page 2, 16 – Comparable to? To previous studies using the 60 day threshold?

- Page 2, 34 – Currently sentinel-2 provides 5-10 day repeat times. Coupled with Landsat, this can provide far more cloud free images of ephemeral snow.

- Page 6, Citation for earth engine

- Noel Gorelick, Matt Hancher, Mike Dixon, Simon Ilyushchenko, David Thau, Rebecca Moore, Google Earth Engine: Planetary-scale geospatial analysis for everyone, Remote Sensing of Environment, Volume 202,

- Figure 1b is not needed, it can be stated in the text.

- Figure 7 - Date ranges for MODIS and SNODAS should be consistent for comparison.

- Figure 8 – Need consistent color bar ranges to aid comparison (or note in caption if you make them different). - Figure 11. 2012 and 2013 "No Snow" look green instead

of black.

---

## Referee Comment (RC2) · Anonymous Referee #2 · 16 Mar 2018

General Comments: This paper addresses an important topic in the hydrology of snow dominated regions. Ephemeral snowpacks are a significant, yet understudied, component of the mountain water balance. This paper identifies key unknowns related to ephemeral snowpacks, presents clear thorough analyses designed to address those unknowns, and concludes recommendations that other investigators can use in future studies. I think the paper falls within the scope of HESS, and is worthy of publication after some moderate revision.

Specific Comments 1. In section 5.1 there is a lot of attention given to lag between date of snow disappearance and date of peak soil moisture in ephemeral vs seasonal
snowpacks. In ephemeral snowpacks the lag times are 79 and 48 days for shallow and deep soil moisture, while in seasonal snowpacks the lag times are about 5 days. However, the actual dates of peak soil moisture are not very different. From figure 5 it appears that the dates of peak soil moisture tends to occur in mid-late may regardless of when the snow disappeared. Does this imply that the timing of snow disappearance in ephemeral snowpacks doesn't really matter to soil moisture? Late winter rain keeps the soil wet in the absence of snow, and peak soil moisture is more a function of the timing of evapotranspiration?

2. The introduction should be modified to better introduce the actual topics in the paper. Specifically, the relationship between ephemeral snowpacks and soil moisture is a dominant them in the paper, but receives little attention in the introduction. Except for a brief mention in the opening paragraph, the term soil moisture doesn't appear again until the research questions in the final paragraph.

3. The writing in some sections needs to be tightened up. Although well organized and generally well-written, it has a feeling of having been written by multiple authors. Section 5.3, for example, has quite a few awkward and complex sentences while other sections are more clear. I suggest a thorough edit of the entire manuscript by a single author.

4. I am not a fan of combined Results and Discussion sections, although I understand the appeal. It is sometimes difficult to decipher what is a result of this study from what is an interpretation of others. Consider separating the sections. This is not a publication deal-breaker, but just something to consider.

Technical Corrections (Page, line) 1,13 Cold content should be defined 1, 32 Is "intermittent" and "ephemeral" the same thing? 4, 8 Goal should be about research questions... 4,15 The soil moisture problem has not been adequately introduced Fig. 2 I don't see the value of this figure. I could be deleted with any impact on the paper if space is a concern 9,11 I don't think these are proper sentences 12,19 Awkward

sentence Fig 8. Panel c has alignment issues 16,2 I don't think the first sentence is necessary. This idea was already introduced. Just start the section with "We propose a..." 16, 5-10 These sentences are redundant with the introduction. They seem out of place in a Results and Discussion section. 16,11 Awkward, complex sentence. I'm not sure what the "based on..." phrase means. Table 1 Average winter temperature estimates should cite the source and method. What duration was used? Probably should round elevations to integer values. Degree symbol is used in caption, but text is used in column heading. 17,6 LaMontagne 2009 is an MS thesis. Better to cite Tyler et al (2008) Tyler, S.W., Burak, S.A., McNamara, J.P., Lamontagne, A., Selker, J.S, and Dozier, J. 2008. Spatially distributed temperatures at the base of two mountain snowpacks measured with fiber optic sensors. Journal of Glaciology, 54(187): 673-679.

Fig 12 Consider putting the years within the figure boxes rather than above them. At first glance, it looks like years should be the x-axis titles.

20, 27 I don't think the problem of defining the length of snow covered periods is an algorithm problem. It's a conceptual understanding, or community definition problem.

---

## Author Comment (AC1) · 7 May 2018

Associate Editor and Reviewers,

We greatly appreciate the thoughtful reviews from the two anonymous reviewers of our paper "Now You See It Now You Don't: A Case Study of Ephemeral Snowpacks and Soil Moisture Response in the Great Basin U.S.A" for publication in Hydrology and Earth System Science. Both reviewers' comments have substantially improved the paper's story and clarity.  We have completed all additional analysis and new figures, with the exception of a seasonal analysis of the SNODAS data for reasons we explain in our responses below in blue text.  Given the changes recommended by the reviewers and our own internal revisions, we believe the revised manuscript has substantial improvements in readability.

We thank the reviewers for recognizing the importance of this under researched topic and its inclusion in to the special edition.  We believe strongly that the submitted manuscript will be of wide interest to HESS readership.

Best regards,

Adrian Harpold

**Anonymous Referee #1**

General comments:

This study focuses on mapping ephemeral snow over the Great Basin in the US and diagnosing the mechanisms for ephemeral snow behavior to better understand the impacts of ephemeral snow on soil moisture. To do so, the authors use station based SNOTEL and SCAN observations of snow and soil moisture, remotely sensed snow-cover from MODIS, and modeled snow data from the SNODAS product. A snow seasonality metric is developed using MODIS snowcover imagery and SNODAS data to map ephemeral and seasonal snowcover. Then a decision tree is developed using SNODAS modeled data to diagnose the mechanism of ephemeral snow. Finally, the landscape characteristics of estimated snow duration are explored. The study's results are that topography and climate are strong controls on the distribution of ephemeral snowpack. This paper extends previous work on ephemeral snow by attempting to classify the processes driving snow loss (no snowfall, melt, sublimation/blowing snow).

This paper addresses a previous unaddressed question of identifying and mapping the dominate process causing ephemeral snow cover. Further, connecting snowmelt from ephemeral snow to soil moisture and this link's sensitivity to inter-annual variability (or climate change), is an interesting scientific issue fully within the scope of HESS. I believe this paper will provide a valuable contribution after some of the issues below are addressed.

Specific comments:

-        Title: The title should be more descriptive of what the paper is about: diagnosing ephemeral snow mechanisms and impacts on soil moisture.

We agree, although we like the visual nature of the previous title and combine to: "Now You See It Now You Don't: A Case Study of Ephemeral Snowpacks and Soil Moisture Response in the Great Basin, USA"

-        The abstract provides some descriptive comments on the seasonality of the observed and modeled results, but only hints at "recommendations to bolster physics based modeling". These recommendations (and results supporting them) should be clearly articulated in the abstract.

We have majorly rewritten the abstract.  The final two sentences address this specific point.

-        Lack of discussion of how uncertainty in the SNODAS model affect the results of this study, namely the classification of ephemeral snow. Over high-elevation terrain where we could expect blowing snow redistribution and sublimation losses to be greatest,

SNODAS at 1km by 1km, likely does not capture these processes well. This may be supported by Figure 7, showing SNODAS diverging from MODIS at highest elevations (This is an interesting finding that could be discussed more as well).

We agree that the finding of mismatches between the SNODAS and MODIS are interesting and worth highlighting. To that end, we have incorporated another figure directly comparable (Fig. 7). The results snow clear biases in the SNODAS estimates of snow duration that we discuss. With regard to the specific point about wind-blown snow effects in SNODAS, we agree and add the following statement at the end of the discussion: "Although SNODAS assimilates MODIS imagery into the model, it does not appear to capture the finer elevation patterns we found using the MOD10A product (Fig. 5 and 6), and in particular, seemed to overestimate consecutive days of snow cover. Part of the challenges at higher elevations is modeling blowing snow patterns over 1-km grid cells, which is consistent lower accuracy of SNODAS above tree line and in more windy areas (Clow et al., 2012; Hedrick et al., 2015). The Great Basin shows tremendous variability in snow ephemerality caused by interactions of topography, elevation, and prevailing wind (Fig. 10-11) and thus, represents an area where improvements in the physically-based modeling will be critical to predicting snow water resources under a variable and changing climate."

It should be noted this statement was already in the text: "Blowing snow sublimation was not the dominant cause of snow ephemerality in the Great Basin for any year, but its known that SNODAS struggles to represent wind redistribution of snow (Clow et al., 2012; Hedrick et al., 2015)."

-        Because ephemeral snow occurs during short events, the driver of snow loss for a given 1km SNODAS cell could be variable with time. How does your ephemeral snow mechanism modeled results change if you look at smaller time slices than a year?

This is an important suggestion, given that the temporal dynamics of ephemeral snow are extreme and seasonal. However, two important methodological concerns kept us from completing this. First is the fidelity of the SNODAS model, which is going to be weaker using a shorter duration window. Second, and more importantly, our maximum consecutive snow duration was developed for an annual time step and is not easily computed (or contextualized) at shorter time steps. The combination of these two concerns makes the execution of this recommendation implausible. However, given the importance of this concern, we add the following sentence: "Our approach to classify proximate causes of snow ephemerality has some limitations. Namely, it assigns only a single mechanism to each grid cell when there could be multiple mechanisms. Moreover, the method cannot consider changes in the mechanisms with time (e.g. melt tends to occur more in spring) because we applied annualized estimates of snow cover duration and concerns about the fidelity of the SNODAS model at short time scales."

-        Snowpillows modify the ground heat flux to snow and the calculated snow presence/absence. Please address how this observational uncertainty impacts your results.

This is a reasonable point and we add the following caveat to the methods: "It should be noted that ablation on the snow pillow may be impacted by differences in ground heat flux and co-location issues with the soil moisture sensors."

- Using the peak of (I assume hourly?) soil moisture data for your calculations for Figure 6, may bias this metric toward high intensity rainfall events (i.e. Feb 2015 in Figure 5e), that may be slightly higher than later snowmelt driven soil moisture increases. Try using a longer averaging time or at least address the sensitivity of your results to this metric choice.

This analysis was done with daily maximum values. We have added a third panel to this figure to help show differences in the absolute (day of year) timing.

- Making your final mapped snow regions publicly available will greatly improve the usefulness of this study.

Yes we agree, we should have this finalized before resubmission.

Technical corrections:

- Page 2, 5 – Missing citation of Kormos et al., 2014

Added

- Page 2, 14 – Inputs "to soil"

Added

- Page 2, 16 – Comparable to? To previous studies using the 60 day threshold?

This sentence was changed to read: "While it is arbitrary, using the 60-day threshold allows for comparisons between the extent of ephemeral snow to previous studies and among different areas."

- Page 2, 34 – Currently sentinel-2 provides 5-10 day repeat times. Coupled with Landsat, this can provide far more cloud free images of ephemeral snow.'

This is worth mentioning and was added to the following sentence: "Given the intermittent nature of ephemeral snow, observations must be daily or finer to capture its dynamics (Wang et al., 2014). Consequently, products like Landsat that has a 16-day overpass and Sentinel that has 5-10 day overpass do poorly at estimating snow seasonality compared to products like the MODIS that have twice daily overpass, but offer untapped potential for merged products with higher spatial and temporal resolution."

We also add this sentence to the recommendations: "While very fine resolution climate datasets are beginning to be produced, there is a large need to merge existing remote sensing snow observations into a data product that maximizes the current space and time resolutions across different platforms (.e.g. spatial resolution of Sentinel 2 but the temporal resolution of MODIS)."

- Page 6, Citation for earth engine

- Noel Gorelick, Matt Hancher, Mike Dixon, Simon Ilyushchenko, David Thau, Rebecca Moore, Google Earth Engine: Planetary-scale geospatial analysis for everyone, Re- mote Sensing of Environment, Volume 202,

This was added

-      Figure 1b is not needed, it can be stated in the text.

We removed this figure.

-      Figure 7 - Date ranges for MODIS and SNODAS should be consistent for comparison.

This was redone for all figures making direct comparisons.

-      Figure 8 – Need consistent color bar ranges to aid comparison (or note in caption if you make them different).

Noted in caption that low elevations have a larger area and require a different caption. But, we took this suggestion to heart and removed the consistent color bars from a later figure.

-      Figure 11. 2012 and 2013 "No Snow" look green instead of black.

We experimented with this and were not happy with the results and do not see a downside of using black.

**Anonymous Referee #2**

General Comments: This paper addresses an important topic in the hydrology of snow dominated regions. Ephemeral snowpacks are a significant, yet understudied, component of the mountain water balance. This paper identifies key unknowns related to ephemeral snowpacks, presents clear thorough analyses designed to address those unknowns, and concludes recommendations that other investigators can use in future studies. I think the paper falls within the scope of HESS, and is worthy of publication after some moderate revision.

Specific Comments 1. In section 5.1 there is a lot of attention given to lag between date of snow disappearance and date of peak soil moisture in ephemeral vs seasonal snowpacks. In ephemeral snowpacks the lag times are 79 and 48 days for shallow and deep soil moisture, while in seasonal snowpacks the lag times are about 5 days. However, the actual dates of peak soil moisture are not very different. From figure 5 it appears that the dates of peak soil moisture tends to occur in mid-late may regardless of when the snow disappeared. Does this imply that the timing of snow disappearance in ephemeral snowpacks doesn't really matter to soil moisture? Late winter rain keeps the soil wet in the absence of snow, and peak soil moisture is more a function of the timing of evapotranspiration?

The consistent timing of deep soil moisture response between ephemeral and seasonal snow zones is a bit of an anomaly due to the lack of deep soil moisture response. We have added a third panel to that figure that shows the absolute timing of peak soil moisture for clarification. The deep soil moisture response in ephemeral areas is strongly biased by the wet years with late snow packs, which are the only years to have sizable deep soil moisture effects (see time series). Given the challenges of interpreting this by the reviewer, we have clarified the text in addition to modifying the figure: "Using similar records to those illustrate at these two sites we use 328 site years (50 ephemeral and 278 seasonal site years) from all SNOTEL and SCAN sites in the Great Basin (Fig. 1) over water year 2014, 2015, and 2016 to illustrate the broader patterns of soil moisture response to ephemeral and seasonal snowmelt. We found that soil moisture following seasonal snowmelt reached a maximum 5 and 7 days prior to snow disappearance for shallow and deep soil moisture, respectively. This confirms previous findings that seasonal snowmelt drives coincident wetting and deeper water percolation (Harpold and Molotch, 2015; McNamara et al., 2005). In contrast, the median soil moisture peaked 79 and 48 days after of snow disappearance from ephemeral snowmelt for shallow and deep soil moisture, respectively (Fig. 4a). This is consistent with the peak shallow soil moisture occurring much earlier in the water year in shallow ephemeral snowmelt areas (Figure 4b). The later deep soil moisture response in ephemeral areas reflects the lack of response, or low coefficient of variation (CV), as compared to seasonal snowmelt (Fig. 4c). The lower CV for deep ephemeral snowmelt (0.2) compared to deep seasonal snowmelt (0.4-0.5) is indicative of reduced deep percolation and less water becoming available to groundwater and streamflow."

2. The introduction should be modified to better introduce the actual topics in the paper. Specifically, the relationship between ephemeral snowpacks and soil moisture is a dominant theme in the paper, but receives little attention in the introduction. Except for a brief mention in the opening paragraph, the term soil moisture doesn't appear again until the research questions in the final paragraph.

This is a great point and was addressed in the revisions. We made several additions and moved information from the discussion to help with clarity.

The second paragraph of the introduction now focuses on soil moisture response: "Snowmelt influences a variety of terrestrial hydrological processes and states,

particularly soil moisture dynamics in areas with low summer precipitation (Harpold and Molotch, 2015; Seyfried et al., 2009). Snowmelt-derived soil moisture is a primary control on streamflow generation and timing and ecosystem productivity in many semi-arid systems (Jefferson, 2011; McNamara et al., 2005; Schwinning and Sala, 2004; Stielstra et al., 2015; Trujillo et al., 2012). Although few studies have isolated their hydrological importance, ephemeral snowpacks modify the intensity and duration of precipitation inputs to soil by storing and releasing water in a less predictable way than seasonal snow. For example, (McNamara et al., 2005) described five predictable phases of soil moisture evolution in semi-arid watersheds with seasonally dominant snowmelt: (1) a summer dry period, (2) a transitional fall wetting period, (3) a winter wet, low-flux period, (4) a spring wet, high-flux period, and (5) a transitional late-spring drying period. Soil moisture response to ephemeral snow melt is likely to sit between the predictable timing and rates of seasonal snow and the stochastic nature of rainfall, but few observations across this gradient exist.  Despite the hydrological and ecological importance of ephemeral snow (McNamara et al., 2018), we lack widely accepted methodologies to classify, map, and model snow ephemerality."

3. The writing in some sections needs to be tightened up. Although well organized and generally well-written, it has a feeling of having been written by multiple authors. Section 5.3, for example, has quite a few awkward and complex sentences while other sections are more clear. I suggest a thorough edit of the entire manuscript by a single author.

- Yes, the senior author has spent time to improve readability.

4. I am not a fan of combined Results and Discussion sections, although I understand the appeal. It is sometimes difficult to decipher what is a result of this study from what is an interpretation of others. Consider separating the sections. This is not a publication deal-breaker, but just something to consider.

I am usually the reviewer making this comment!  I agree that combined results/discussion is often not the ideal format for a results heavy article, however, in our case this format allows us to link three generally disparate ideas (soil moisture, MODIS patterns, and SNODAS mechanisms) into a comprehensive story.  Because this is the first broad paper about ephemeral snow hydrology, the broader story is more conducive to a combined results and discussion.  We hope you find the new version easy to digest.

Technical Corrections (Page, line) 1,13 Cold content should be defined

1, 32 Is "in- termittent" and "ephemeral" the same thing?

Yes and this was clarified in the text.

4, 8 Goal should be about research questions. . .

We agree and have modified this sentence to: "The goal of this paper is to use the Great Basin as a case study to estimate the distribution and mechanisms causing ephemeral snow to better constrain their impact on soil moisture and hydrological response."

4,15 The soil moisture problem has not been adequately introduced

This has been changed. See previous comments.

Fig. 2 I don't see the value of this figure. I could be deleted with any impact on the paper if space is a concern

We moved Figure 2 and 3 to the supplemental.

9,11 I don't think these are proper sentences

12,19 Awkward sentence reen

Changed to read.

Fig 8. Panel c has alignment issues

This has been fixed

16,2 I don't think the first sentence is necessary. This idea was already introduced. Just start the section with "We propose a. . ."

Agree this change was made.

16, 5-10 These sentences are redundant with the introduction. They seem out of place in a Results and Discussion section.

Agreed. This was removed and some text moved to other sections.

6,11 Awkward, complex sentence. I'm not sure what the "based on. . ." phrase means.

This was changed.

Table 1 Average winter temperature estimates should cite the source and method. What duration was used? Probably should round elevations to integer values. Degree symbol is used in caption, but text is used in column heading.

This was clarified to be December 1 to April 1 in all locations. We have added the degree symbol in all instances.

17,6 LaMontagne 2009 is an MS thesis. Better to cite Tyler et al (2008) Tyler, S.W., Burak, S.A., McNamara, J.P., Lamontagne, A., Selker, J.S, and Dozier, J. 2008. Spatially

distributed temperatures at the base of two mountain snow- packs measured with fiber optic sensors. Journal of Glaciology, 54(187): 673-679.

Thank you for this. We also added the reference: Slater, A. G., D. M. Lawrence, and C. D. Koven (2017), Process-level model evaluation: a snow and heat transfer metric, *The Cryosphere*, *11*(2), 989–996, doi:10.5194/tc-11-989-2017.

Fig 12 Consider putting the years within the figure boxes rather than above them. At first glance, it looks like years should be the x-axis titles.

This was changed.

20, 27 I don't think the problem of defining the length of snow covered periods is an algorithm problem. It's a conceptual understanding, or community definition problem.

We agree it's a definition problem, but it is also an algorithm problem. We have clarified this to read: "
[revised manuscript text omitted]

---

## Editor Comment (EC1) · S. Carey (Editor) · 31 Jul 2018

Dear Authors

I would like to thank you for the time and care that has gone in to the revised manuscript and for detailing your response to the reviewers comments. The issue of ephemeral snow is a vexing one, and indeed one that is not limited to mountain basins where the rain/snow transition is changing. The snow literature in hydrology is strongly biased towards deep mountain snowpacks and 'water tower' systems with less emphasis on the accumulation/melt cycle that dominates many environments that rise and fall below freezing throughout winter. I would particularly like to note the efforts you have made

to outline new metrics to define ephemeral snow systems, and highlight the factors that control this process in the Great Basin.

There are a number of small revisions that I would like you to consider in a final revised manuscript, which I believe is suitable for publication in the HESS Special Issue on Understanding and predicting Earth system and hydrological change in cold regions.

Figure 1. The labels on the right are very large and I believe the figure would improve if they were smaller. In addition, State boundaries would help the reader locate the Great Basin within the United States. In a number of the later figures, the geographical extent changes (Figure 10 for example). I believe a standardized map domain with State boundaries (or a larger boundary for context) would help non US readers. In later figures, labels are also very large.

On page 15, GRIDMET is used as an acronym but this product needs to be explained in more detail.

Page 15, line 23. Change its to it is, or simply state "for any year, SNODAS struggles to..."

Table 1. Elevation probably does not need a sub-meter resolution.

Finally, the manuscript uses a lot of first-person viewpoints. "We" was used 66 times in the manuscript and I believe that with some simple editing, could be reduced.

Sean Carey McMaster University

---

## Author Response (AR3)

Associate Editor and Reviewers,

We greatly appreciate the thoughtful reviews from the two anonymous reviewers of our paper "Now You See It Now You Don't: A Case Study of Ephemeral Snowpacks and Soil Moisture Response in the Great Basin U.S.A" for publication in Hydrology and Earth System Science. Both reviewers' comments have substantially improved the paper's story and clarity.  We have completed all additional analysis and new figures, with the exception of a seasonal analysis of the SNODAS data for reasons we explain in our responses below in blue text.  Given the changes recommended by the reviewers and our own internal revisions, we believe the revised manuscript has substantial improvements in readability.

We thank the reviewers for recognizing the importance of this under researched topic and its inclusion in to the special edition.  We believe strongly that the submitted manuscript will be of wide interest to HESS readership.

Best regards,

Adrian Harpold

Dr. Sean Carey,

We, the authors of *Now You See It Now You Don't: A Case Study of Ephemeral Snowpacks and Soil Moisture Response in the Great Basin U.S.A.* appreciate your involvement in the process of publication and your suggestions about improving our final draft.

On Figure 1, we have added state boundaries per your suggestion. We have also shrunk the labeling and legend sizes in Figures 1, 5, 10 and 11.

The sentence about GRIDMET now reads: "In that six-year period, the year with the lowest average winter (Dec 1st to Apr 1st) temperature using gridded meterological (GRIDMET) 4 km resolution surface temperature estimates was 2013 at -0.9 °C while the year with the highest average winter temperature was 2014 at 1.0 °C (Abatzoglou 2012; Table 1)" (P16L12). which adds information about the resolution of GRIDMET along with clarifying that it is used to estimate surface temperature.

The grammatical mistake that you pointed out when discussing the absence of blowing snow in our results was omitted, and we have revised the sentence at P16L22 to read: "Blowing snow sublimation was not the dominant cause of snow ephemerality in the Great Basin for any year; SNODAS struggles to represent wind redistribution of snow (Clow et al., 2012; Hedrick et al., 2015)." This is very similar to your suggestion with the one difference being that we used a semicolon in the place of a comma in order to avoid a comma splice.

We have removed the submeter resolution on elevation in Table 1.

Lastly, we appreciate your comment on the use of "we" in our manuscript and have reduced their number to 29.

Thank you for recognizing the importance of ephemeral snow in the field of snow hydrology, and for selecting our paper for the *Hydrology and Earth Systems Science* special edition.

Sincerely yours,

Rose Petersky

Adrian Harpold

**Anonymous Referee #1**

General comments:

This study focuses on mapping ephemeral snow over the Great Basin in the US and diagnosing the mechanisms for ephemeral snow behavior to better understand the impacts of ephemeral snow on soil moisture. To do so, the authors use station based SNOTEL and SCAN observations of snow and soil moisture, remotely sensed snow-cover from MODIS, and modeled snow data from the SNODAS product. A snow seasonality metric is developed using MODIS snowcover imagery and SNODAS data to map ephemeral and seasonal snowcover. Then a decision tree is developed using SNODAS modeled data to diagnose the mechanism of ephemeral snow. Finally, the landscape characteristics of estimated snow duration are explored. The study's results are that topography and climate are strong controls on the distribution of ephemeral snowpack. This paper extends previous work on ephemeral snow by attempting to classify the processes driving snow loss (no snowfall, melt, sublimation/blowing snow).

This paper addresses a previous unaddressed question of identifying and mapping the dominate process causing ephemeral snow cover. Further, connecting snowmelt from ephemeral snow to soil moisture and this link's sensitivity to inter-annual variability (or climate change), is an interesting scientific issue fully within the scope of HESS. I believe this paper will provide a valuable contribution after some of the issues below are addressed.

Specific comments:

-       Title: The title should be more descriptive of what the paper is about: diagnosing ephemeral snow mechanisms and impacts on soil moisture.

We agree, although we like the visual nature of the previous title and combine to: "Now You See It Now You Don't: A Case Study of Ephemeral Snowpacks and Soil Moisture Response in the Great Basin, USA"

-       The abstract provides some descriptive comments on the seasonality of the observed and modeled results, but only hints at "recommendations to bolster physics based modeling". These recommendations (and results supporting them) should be clearly articulated in the abstract.

We have majorly rewritten the abstract.  The final two sentences address this specific point.

-       Lack of discussion of how uncertainty in the SNODAS model affect the results of this study, namely the classification of ephemeral snow. Over high-elevation terrain where we could expect blowing snow redistribution and sublimation losses to be greatest,

SNODAS at 1km by 1km, likely does not capture these processes well. This may be supported by Figure 7, showing SNODAS diverging from MODIS at highest elevations (This is an interesting finding that could be discussed more as well).

We agree that the finding of mismatches between the SNODAS and MODIS are interesting and worth highlighting. To that end, we have incorporated another figure directly comparable (Fig. 7). The results snow clear biases in the SNODAS estimates of snow duration that we discuss. With regard to the specific point about wind-blown snow effects in SNODAS, we agree and add the following statement at the end of the discussion: "Although SNODAS assimilates MODIS imagery into the model, it does not appear to capture the finer elevation patterns we found using the MOD10A product (Fig. 5 and 6), and in particular, seemed to overestimate consecutive days of snow cover. Part of the challenges at higher elevations is modeling blowing snow patterns over 1-km grid cells, which is consistent lower accuracy of SNODAS above tree line and in more windy areas (Clow et al., 2012; Hedrick et al., 2015). The Great Basin shows tremendous variability in snow ephemerality caused by interactions of topography, elevation, and prevailing wind (Fig. 10-11) and thus, represents an area where improvements in the physically-based modeling will be critical to predicting snow water resources under a variable and changing climate."

It should be noted this statement was already in the text: "Blowing snow sublimation was not the dominant cause of snow ephemerality in the Great Basin for any year, but its known that SNODAS struggles to represent wind redistribution of snow (Clow et al., 2012; Hedrick et al., 2015)."

-        Because ephemeral snow occurs during short events, the driver of snow loss for a given 1km SNODAS cell could be variable with time. How does your ephemeral snow mechanism modeled results change if you look at smaller time slices than a year?

This is an important suggestion, given that the temporal dynamics of ephemeral snow are extreme and seasonal. However, two important methodological concerns kept us from completing this. First is the fidelity of the SNODAS model, which is going to be weaker using a shorter duration window. Second, and more importantly, our maximum consecutive snow duration was developed for an annual time step and is not easily computed (or contextualized) at shorter time steps. The combination of these two concerns makes the execution of this recommendation implausible. However, given the importance of this concern, we add the following sentence: "Our approach to classify proximate causes of snow ephemerality has some limitations. Namely, it assigns only a single mechanism to each grid cell when there could be multiple mechanisms. Moreover, the method cannot consider changes in the mechanisms with time (e.g. melt tends to occur more in spring) because we applied annualized estimates of snow cover duration and concerns about the fidelity of the SNODAS model at short time scales."

-        Snowpillows modify the ground heat flux to snow and the calculated snow presence/absence. Please address how this observational uncertainty impacts your results.

This is a reasonable point and we add the following caveat to the methods: "It should be noted that ablation on the snow pillow may be impacted by differences in ground heat flux and co-location issues with the soil moisture sensors."

- Using the peak of (I assume hourly?) soil moisture data for your calculations for Figure 6, may bias this metric toward high intensity rainfall events (i.e. Feb 2015 in Figure 5e), that may be slightly higher than later snowmelt driven soil moisture increases. Try using a longer averaging time or at least address the sensitivity of your results to this metric choice.

This analysis was done with daily maximum values.  We have added a third panel to this figure to help show differences in the absolute (day of year) timing.

- Making your final mapped snow regions publicly available will greatly improve the usefulness of this study.

Yes we agree, we should have this finalized before resubmission.

Technical corrections:

- Page 2, 5 – Missing citation of Kormos et al., 2014

Added

- Page 2, 14 – Inputs "to soil"

Added

- Page 2, 16 – Comparable to? To previous studies using the 60 day threshold?

This sentence was changed to read: "While it is arbitrary, using the 60-day threshold allows for comparisons between the extent of ephemeral snow to previous studies and among different areas."

- Page 2, 34 – Currently sentinel-2 provides 5-10 day repeat times. Coupled with Landsat, this can provide far more cloud free images of ephemeral snow.'

This is worth mentioning and was added to the following sentence: "Given the intermittent nature of ephemeral snow, observations must be daily or finer to capture its dynamics (Wang et al., 2014). Consequently, products like Landsat that has a 16-day overpass and Sentinel that has 5-10 day overpass do poorly at estimating snow seasonality compared to products like the MODIS that have twice daily overpass, but offer untapped potential for merged products with higher spatial and temporal resolution."

We also add this sentence to the recommendations: "While very fine resolution climate datasets are beginning to be produced, there is a large need to merge existing remote sensing snow observations into a data product that maximizes the current space and time resolutions across different platforms (.e.g. spatial resolution of Sentinel 2 but the temporal resolution of MODIS)."

- Page 6, Citation for earth engine

- Noel Gorelick, Matt Hancher, Mike Dixon, Simon Ilyushchenko, David Thau, Rebecca Moore, Google Earth Engine: Planetary-scale geospatial analysis for everyone, Re- mote Sensing of Environment, Volume 202,

This was added

-       Figure 1b is not needed, it can be stated in the text.

We removed this figure.

-       Figure 7 - Date ranges for MODIS and SNODAS should be consistent for comparison.

This was redone for all figures making direct comparisons.

-       Figure 8 – Need consistent color bar ranges to aid comparison (or note in caption if you make them different).

Noted in caption that low elevations have a larger area and require a different caption. But, we took this suggestion to heart and removed the consistent color bars from a later figure.

-       Figure 11. 2012 and 2013 "No Snow" look green instead of black.

We experimented with this and were not happy with the results and do not see a downside of using black.

**Anonymous Referee #2**

General Comments: This paper addresses an important topic in the hydrology of snow dominated regions. Ephemeral snowpacks are a significant, yet understudied, component of the mountain water balance. This paper identifies key unknowns related to ephemeral snowpacks, presents clear thorough analyses designed to address those unknowns, and concludes recommendations that other investigators can use in future studies. I think the paper falls within the scope of HESS, and is worthy of publication after some moderate revision.

Specific Comments 1. In section 5.1 there is a lot of attention given to lag between date of snow disappearance and date of peak soil moisture in ephemeral vs seasonal snowpacks. In ephemeral snowpacks the lag times are 79 and 48 days for shallow and deep soil moisture, while in seasonal snowpacks the lag times are about 5 days. However, the actual dates of peak soil moisture are not very different. From figure 5 it appears that the dates of peak soil moisture tends to occur in mid-late may regardless of when the snow disappeared. Does this imply that the timing of snow disappearance in ephemeral snowpacks doesn't really matter to soil moisture? Late winter rain keeps the soil wet in the absence of snow, and peak soil moisture is more a function of the timing of evapotranspiration?

The consistent timing of deep soil moisture response between ephemeral and seasonal snow zones is a bit of an anomaly due to the lack of deep soil moisture response.  We have added a third panel to that figure that shows the absolute timing of peak soil moisture for clarification.  The deep soil moisture response in ephemeral areas is strongly biased by the wet years with late snow packs, which are the only years to have sizable deep soil moisture effects (see time series).  Given the challenges of interpreting this by the reviewer, we have clarified the text in addition to modifying the figure: "Using similar records to those illustrate at these two sites we use 328 site years (50 ephemeral and 278 seasonal site years) from all SNOTEL and SCAN sites in the Great Basin (Fig. 1) over water year 2014, 2015, and 2016 to illustrate the broader patterns of soil moisture response to ephemeral and seasonal snowmelt. We found that soil moisture following seasonal snowmelt reached a maximum 5 and 7 days prior to snow disappearance for shallow and deep soil moisture, respectively. This confirms previous findings that seasonal snowmelt drives coincident wetting and deeper water percolation (Harpold and Molotch, 2015; McNamara et al., 2005). In contrast, the median soil moisture peaked 79 and 48 days after of snow disappearance from ephemeral snowmelt for shallow and deep soil moisture, respectively (Fig. 4a). This is consistent with the peak shallow soil moisture occurring much earlier in the water year in shallow ephemeral snowmelt areas (Figure 4b). The later deep soil moisture response in ephemeral areas reflects the lack of response, or low coefficient of variation (CV), as compared to seasonal snowmelt (Fig. 4c). The lower CV for deep ephemeral snowmelt (0.2) compared to deep seasonal snowmelt (0.4-0.5) is indicative of reduced deep percolation and less water becoming available to groundwater and streamflow."

2. The introduction should be modified to better introduce the actual topics in the paper. Specifically, the relationship between ephemeral snowpacks and soil moisture is a dominant theme in the paper, but receives little attention in the introduction. Except for a brief mention in the opening paragraph, the term soil moisture doesn't appear again until the research questions in the final paragraph.

This is a great point and was addressed in the revisions.  We made several additions and moved information from the discussion to help with clarity.

The second paragraph of the introduction now focuses on soil moisture response: "Snowmelt influences a variety of terrestrial hydrological processes and states,

particularly soil moisture dynamics in areas with low summer precipitation (Harpold and Molotch, 2015; Seyfried et al., 2009). Snowmelt-derived soil moisture is a primary control on streamflow generation and timing and ecosystem productivity in many semi-arid systems (Jefferson, 2011; McNamara et al., 2005; Schwinning and Sala, 2004; Stielstra et al., 2015; Trujillo et al., 2012). Although few studies have isolated their hydrological importance, ephemeral snowpacks modify the intensity and duration of precipitation inputs to soil by storing and releasing water in a less predictable way than seasonal snow. For example, (McNamara et al., 2005) described five predictable phases of soil moisture evolution in semi-arid watersheds with seasonally dominant snowmelt: (1) a summer dry period, (2) a transitional fall wetting period, (3) a winter wet, low-flux period, (4) a spring wet, high-flux period, and (5) a transitional late-spring drying period. Soil moisture response to ephemeral snow melt is likely to sit between the predictable timing and rates of seasonal snow and the stochastic nature of rainfall, but few observations across this gradient exist. Despite the hydrological and ecological importance of ephemeral snow (McNamara et al., 2018), we lack widely accepted methodologies to classify, map, and model snow ephemerality."

3. The writing in some sections needs to be tightened up. Although well organized and generally well-written, it has a feeling of having been written by multiple authors. Section 5.3, for example, has quite a few awkward and complex sentences while other sections are more clear. I suggest a thorough edit of the entire manuscript by a single author.

- Yes, the senior author has spent time to improve readability.

4. I am not a fan of combined Results and Discussion sections, although I understand the appeal. It is sometimes difficult to decipher what is a result of this study from what is an interpretation of others. Consider separating the sections. This is not a publication deal-breaker, but just something to consider.

I am usually the reviewer making this comment!  I agree that combined results/discussion is often not the ideal format for a results heavy article, however, in our case this format allows us to link three generally disparate ideas (soil moisture, MODIS patterns, and SNODAS mechanisms) into a comprehensive story.  Because this is the first broad paper about ephemeral snow hydrology, the broader story is more conducive to a combined results and discussion.  We hope you find the new version easy to digest.

Technical Corrections (Page, line) 1,13 Cold content should be defined

1, 32 Is "in- termittent" and "ephemeral" the same thing?

Yes and this was clarified in the text.

4, 8 Goal should be about research questions. . .

We agree and have modified this sentence to: "The goal of this paper is to use the Great Basin as a case study to estimate the distribution and mechanisms causing ephemeral snow to better constrain their impact on soil moisture and hydrological response."

4,15 The soil moisture problem has not been adequately introduced

This has been changed. See previous comments.

Fig. 2 I don't see the value of this figure. I could be deleted with any impact on the paper if space is a concern

We moved Figure 2 and 3 to the supplemental.

9,11 I don't think these are proper sentences

12,19 Awkward sentence reen

Changed to read.

Fig 8. Panel c has alignment issues

This has been fixed

16,2 I don't think the first sentence is necessary. This idea was already introduced. Just start the section with "We propose a. . ."

Agree this change was made.

16, 5-10 These sentences are redundant with the introduction. They seem out of place in a Results and Discussion section.

Agreed. This was removed and some text moved to other sections.

6,11 Awkward, complex sentence. I'm not sure what the "based on. . ." phrase means.

This was changed.

Table 1 Average winter temperature estimates should cite the source and method. What duration was used? Probably should round elevations to integer values. Degree symbol is used in caption, but text is used in column heading.

This was clarified to be December 1 to April 1 in all locations. We have added the degree symbol in all instances.

17,6 LaMontagne 2009 is an MS thesis. Better to cite Tyler et al (2008) Tyler, S.W., Burak, S.A., McNamara, J.P., Lamontagne, A., Selker, J.S, and Dozier, J. 2008. Spatially

distributed temperatures at the base of two mountain snow- packs measured with fiber optic sensors. Journal of Glaciology, 54(187): 673-679.

Thank you for this. We also added the reference: Slater, A. G., D. M. Lawrence, and C. D. Koven (2017), Process-level model evaluation: a snow and heat transfer metric, *The Cryosphere*, *11*(2), 989–996, doi:10.5194/tc-11-989-2017.

Fig 12 Consider putting the years within the figure boxes rather than above them. At first glance, it looks like years should be the x-axis titles.

This was changed.

20, 27 I don't think the problem of defining the length of snow covered periods is an algorithm problem. It's a conceptual understanding, or community definition problem.

We agree it's a definition problem, but it is also an algorithm problem. We have clarified this to read: "
[revised manuscript text omitted]

In additionUsing similar records to comparing soil moisture responses forthose illustrated at these two sites, we also analyzeduse 328 site years (50 ephemeral and 278 seasonal site years) from all SNOTEL and SCAN sites in the Great Basin (Fig. 1) over water yearyears 2014, 2015, and 2016 in order to illustrate the broader patterns of soil moisture betweenresponse to ephemeral and seasonal snow meltsnowmelt. We found that soil moisture following seasonal snow melt peaked on averagesnowmelt reached a maximum 5 and 7 days prior to snow disappearance for shallow and deep soil moisture, respectively. This confirms previous findings that seasonal snow meltsnowmelt drives coincident wetting and deeper water percolation (Harpold and Molotch, 2015; McNamara et al., 2005). In contrast, the median soil moisture peaked 79 and 48 days after of snow disappearance from ephemeral snow meltsnowmelt for shallow and deep soil moisture, respectively (Fig. 6a). Deep4a). This is consistent with the peak shallow soil moisture occurring much earlier in the water year in shallow ephemeral snowmelt had aareas (Fig. 4b). The later deep soil moisture response in ephemeral areas reflects the lack of response, or low coefficient of variation (CV) of 0.2), as compared to 0.4-0.5 forseasonal snowmelt (Fig. 64c). The lower CV for deep ephemeral snowmelt (0.2) compared to deep seasonal snow melt likely reflectssnowmelt (0.4-0.5) is indicative of reduced deep percolation and less water becoming available to groundwater and streamflow.

[Figure]

**Figure 4:** (a) The difference between date of peak soil moisture and last day of snow (Days) for shallow (5 cm) and deep (50 cm) soil moisture during water years 2014-2016 in Great Basin SNOTEL stations with ephemeral snow (50 site years) and seasonal snow (278 site years). (b) Day of peak soil moisture for SNOTEL and SCAN stations for shallow (5 cm) and deep (50 cm) soil moisture during water years 2014-2016 (c) The coefficient of variation (CV) for shallow (5 cm) and deep (50 cm) soil moisture during water years 2014-2016

The differences in soil moisture response between seasonal and ephemeral snowpacks across the Great Basin could have important consequences for vegetation phenology and runoff generation. For example, the timing of soil moisture is a strong control on the timing and amount of net ecosystem productivity (Inouye, 2008), with earlier snowmelt causing an earlier and longer growing season with reduced carbon uptake (Hu et al., 2010; Winchell et al., 2016). (Harpold, 2016) also showed that earlier snow disappearance generally led to more days of soil moisture below wilting point at SNOTEL sites. Our finding that soil moisture peaked earlier in ephemeral snowmelt than seasonal snowmelt is thus likely to be correlated with reduced vegetation productivity and increased late season water stress in many areas. In addition to stressing local vegetation, ephemeral snowmelt may reduce groundwater recharge and streamflow. For example, baseflow contributions to streamflow and overall water yield declined when snowmelt rates were smaller (Barnhart et al., 2016; Earman et al., 2006; Trujillo and Molotch, 2014) and overall water yields were lower in basins receiving more rain and less snow (Berghuijs et al., 2014). Changes in percolation patterns also affect the distribution of more shallow rooting plants versus deeper rooting plants that need long duration soil moisture pulses to grow and reproduce (Schwinning and Sala, 2004). These differences in how ephemeral versus seasonal snowmelt   soil moisture provide a strong motivation to understand the distribution and causes of ephemeral snowpacks across the Great Basin.

**4.2 Topographic Controls on Snow Seasonality**

[revised manuscript text omitted]